# System-Embedded Diffusion Bridge Models

**Bartlomiej Sobieski**[*†1]**, Matthew Tivnan**[†2,3]**, Yuang Wang**[2,3]**, Siyeop Yoon**[2,3]**,**
**Pengfei Jin**[2,3]**, Dufan Wu**[2,3]**, Quanzheng Li**[2,3]**, Przemyslaw Biecek**[1,4]
[1]University of Warsaw, [2]Harvard University,
[3]Massachusetts General Hospital, [4]Warsaw University of Technology

## Abstract

Solving inverse problems—recovering signals from incomplete or noisy measurements—is fundamental in science and engineering. Score-based generative models (SGMs) have recently emerged as a powerful framework for this task. Two main paradigms have formed: unsupervised approaches that adapt pretrained generative models to inverse problems, and supervised bridge methods that train stochastic processes conditioned on paired clean and corrupted data. While the former typically assume knowledge of the measurement model, the latter have largely overlooked this structural information. We introduce System-embedded Diffusion Bridge Models (SDBs), a new class of supervised bridge methods that explicitly embed the known linear measurement system into the coefficients of a matrix-valued SDE. This principled integration yields consistent improvements across diverse linear inverse problems and demonstrates robust generalization under system misspecification between training and deployment, offering a promising solution to real-world applications.

## 1 Introduction

Restoring data from corrupted or incomplete measurements is a fundamental task in science and engineering [Abbott et al., 2016, Akiyama et al., 2019], commonly referred to as an *inverse problem*. Its *linear* formulation plays a central role in practical domains such as signal processing and medical imaging [Bertero et al., 1985, Candes et al., 2006]. The emergence of deep learning has significantly advanced this field, enabling major scientific breakthroughs [Ongie et al., 2020]. Since the work of Song et al. [2021b], the scientific community has increasingly adopted *score-based generative models* (SGMs), also known as *diffusion models*, to tackle inverse problems. Two key directions have emerged: one adapts models pretrained for image synthesis to conditional generation; the other, often called *bridge methods*, trains problem-specific models grounded in stochastic differential equations (SDEs), assuming access to paired samples of clean data and corresponding measurements.

While pretrained models often assume access to a known linear measurement system, bridge methods have primarily focused on developing general-purpose approaches without leveraging such structural information. However, in many real-world settings—such as CT or MRI reconstruction—the linear measurement process is known *a priori*, and datasets frequently contain paired examples of clean and degraded data.

To address this gap, we propose *System-embedded Diffusion Bridge Models* (SDBs), a novel method that incorporates the system response and noise covariance directly into the coefficients of a matrix-valued SDE. By embedding this measurement-system knowledge, SDB achieves considerable performance gains across diverse linear systems of varying complexity, demonstrated through three distinct instantiations. Furthermore, we conduct an extensive study of SDB's generalization under system

---

[*]Corresponding author at `b.sobieski@uw.edu.pl`
[†]Equal contribution

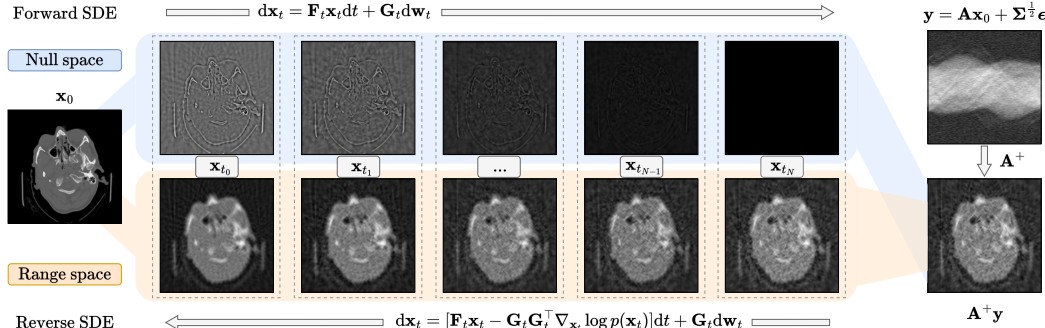

Figure 1: SDB learns a diffusion bridge between pseudoinverse reconstructions and clean samples by embedding the measurement system into the coefficients of the linear SDE, allowing the score model to distinguish between the range and null spaces of the task.

misspecification between training and deployment, showing that SDB consistently outperforms baselines and remains robust in the face of parameter shifts, making it well-suited for real-world deployment.

In addition, our work advocates for more expressive formulations of diffusion processes in generative modeling, opening avenues for future research into system-aware and structure-driven stochastic modeling frameworks.[3]

## 2 Related works

Our work offers a new perspective on constructing task-specific diffusion bridges (*supervised*), while drawing inspiration from diffusion-based plug-and-play methods (*unsupervised*). Both directions address the topic of image restoration and inverse problems from different viewpoints. Below, we briefly cover these directions, with a much more detailed overview included in the Appendix.

**Diffusion models for inverse problems.**    As shown in the work of Song et al. [2021b], pretrained unconditional diffusion models can be easily adapted to conditional synthesis with a simple application of Bayes' Theorem. The resulting likelihood term has been extensively utilized for guided generation [Dhariwal and Nichol, 2021], especially for image restoration problems [Daras et al., 2024]. These methods range from generic ones [Song et al., 2021b, Chung et al., 2022] to those that incorporate additional assumptions and constraints to the generation process, such as DPS [Chung et al., 2023a], DDNM [Wang et al., 2023], ΠGDM [Song et al., 2023a], DDPG [Garber and Tirer, 2024], and others [Kawar et al., 2022, Chung et al., 2022, 2023a, Song et al., 2023a, Mardani et al., 2024, Chung et al., 2024].

**Diffusion bridges.**    One may also extend the SGM framework to arrive at mappings between arbitrary probability distributions given paired data [Särkkä and Solin, 2019]. Among many seminal works in this area [Heng et al., 2021, Somnath et al., 2023, Peluchetti, 2022, Delbracio and Milanfar, 2023, De Bortoli et al., 2021], Liu et al. [2023] (I2SB) propose a simulation-free algorithm that solves the Schrödinger Bridge (SB) problem, Luo et al. [2023a] (IR-SDE) construct a conditional stochastic differential equation (SDE) that maps noisy degradations to clean samples, while Yue et al. [2024] (GOUB) generalize it with the Doob's h-transform [Doob, 1984], which is also used by Zhou et al. [2024] (DDBM) to generalize the entire SGM framework.

## 3 Background

**Diffusion models.**    Generative modeling of visual data has advanced rapidly with the introduction of diffusion models [Sohl-Dickstein et al., 2015, Ho et al., 2020, Dhariwal and Nichol, 2021], which

---

[3]We include the source code at https://github.com/sobieskibj/sdb.

generate images by sequential denoising. Initially formulated as a discrete Markov chain with Gaussian kernels, diffusion models were later unified with score matching [Song and Ermon, 2019] under the framework of SDEs [Song et al., 2021b], establishing the continuous-time formulation known as SGM.

Let $p(\mathbf{x}_t)$ denote the probability distribution of $\mathbf{x}_t \in \mathbb{R}^d$ parameterized by time $t \in [0, 1]$, where $p(\mathbf{x}_0)$ represents the data distribution and $p(\mathbf{x}_1)$ the prior. In this work, we highlight the often-overlooked *linear* formulation of SGM. The forward SDE, which maps $p(\mathbf{x}_0)$ to $p(\mathbf{x}_1)$, is defined as

$$\mathrm{d}\mathbf{x}_t = \mathbf{F}_t\mathbf{x}_t\mathrm{d}t + \mathbf{G}_t\mathrm{d}\mathbf{w}_t, \tag{1}$$

where $\mathbf{F}_t\mathbf{x}_t$ is a linear drift term with time-dependent matrix $\mathbf{F}_t \in \mathbb{R}^{d \times d}$, and $\mathbf{G}_t$ is a matrix-valued diffusion coefficient. Following Anderson [1982], the reverse SDE, which maps $p(\mathbf{x}_1)$ to $p(\mathbf{x}_0)$, is given by

$$\mathrm{d}\mathbf{x}_t = [\mathbf{F}_t\mathbf{x}_t - \mathbf{G}_t\mathbf{G}_t^\top \nabla_{\mathbf{x}_t}\log p(\mathbf{x}_t)]\mathrm{d}t + \mathbf{G}_t\mathrm{d}\overline{\mathbf{w}}_t, \tag{2}$$

where $\nabla_{\mathbf{x}_t}\log p(\mathbf{x}_t)$ is the score function and $\overline{\mathbf{w}}_t$ is the Wiener process with reversed time. The forward-reverse relationship ensures that the marginal distributions $p(\mathbf{x}_t)$, defined by eq. (1) and eq. (2), remain consistent.

In standard SGM for image synthesis, $p(\mathbf{x}_1)$ is typically chosen as an isotropic Gaussian. To generate samples from $p(\mathbf{x}_0)$, one samples from $p(\mathbf{x}_1)$ and solves eq. (2) using a time discretization scheme. This requires the unknown score function $\nabla_{\mathbf{x}_t}\log p(\mathbf{x}_t)$, which is generally approximated by a neural network $\mathbf{s}_{\boldsymbol{\theta}}(\mathbf{x}_t, t)$ trained with the score-matching loss [Song et al., 2021b]:

$$\mathbb{E}[|\mathbf{s}_{\boldsymbol{\theta}}(\mathbf{x}_t, t) - \nabla_{\mathbf{x}_t}\log p(\mathbf{x}_t \mid \mathbf{x}_0)|_2^2], \tag{3}$$

where the expectation is taken over $\mathbf{x}_0 \sim p(\mathbf{x}_0)$, $t \sim \mathcal{U}([0, 1])$, and $\mathbf{x}_t \sim p(\mathbf{x}_t \mid \mathbf{x}_0)$.

The linear formulation in eq. (1) and eq. (2) is a special case of the broader SGM framework introduced by Song et al. [2021b], where drift and diffusion coefficients can be arbitrarily complex functions of $\mathbf{x}_t$ and $t$. This linear formulation, however, generalizes the well-established *scalar* case. Specifically, the Variance-Preserving (VP) SDE, which maintains constant variance for $\mathbf{x}_t$ across time, can be recovered with $\mathbf{F_t} = -\frac{1}{2}\beta_t\mathbf{I}$ and $\mathbf{G}_t = \sqrt{\beta_t}\mathbf{I}$. The Variance-Exploding (VE) SDE, which induces unbounded growth in the variance of $p(\mathbf{x}_1)$ as $t \to 1$, is obtained with $\mathbf{F_t} = \mathbf{0}$ and $\mathbf{G}_t = \sqrt{\frac{\mathrm{d}\sigma_t^2}{\mathrm{d}t}}\mathbf{I}$. While these scalar matrices offer simplicity, they also raise the question of whether more sophisticated designs could yield additional benefits.

**Inverse problems.** For a signal sample $\mathbf{x} \in \mathbb{R}^d$, we define the *linear measurement system* as

$$\mathbf{y} = \mathbf{A}\mathbf{x} + \boldsymbol{\Sigma}^{\frac{1}{2}}\boldsymbol{\epsilon}, \tag{4}$$

where $\mathbf{y} \in \mathbb{R}^m$ is the *measurement*, $\boldsymbol{A} \in \mathbb{R}^{m \times d}$ is the *system response matrix*, $\boldsymbol{\epsilon} \sim \mathcal{N}(\mathbf{0}_m, \mathbf{I}_{m \times m})$ is the measurement noise with covariance matrix $\boldsymbol{\Sigma} \in \mathbb{R}^{m \times m}$ and $p(\mathbf{y} \mid \mathbf{x}) = \mathcal{N}(\mathbf{A}\mathbf{x}, \boldsymbol{\Sigma})$ is the *forward model*. We refer to finding the unknown $\mathbf{x}$ based on the provided $\mathbf{y}$ as solving a linear inverse problem with additive Gaussian noise [Tarantola, 2005]. This formulation is central to many practical applications, such as signal processing, medical imaging, and remote sensing [Mueller and Siltanen, 2012], particularly in the case of noninvertible linear systems, where either $m \neq d$ or $\mathbf{A}$ is rank-deficient.

**Matrix pseudoinverse.** Consider the simplest noiseless ($\boldsymbol{\Sigma} = \mathbf{0}_{m \times m}$) setting of eq. (4) with invertible $\mathbf{A}$. In this case, the original signal $\mathbf{x}$ can be fully recovered by simply using the inverse of $\mathbf{A}$, *i.e.*, $\mathbf{x} = \mathbf{A}^{-1}\mathbf{y}$. However, this is only possible when $m = d$ and $\mathbf{A}$ is full-rank, which greatly limits the scope of potential inverse problems. In practice, the matrix $\mathbf{A}$ is typically not invertible, and noise arises naturally due to, *e.g.*, imperfections in the measurement process.

In the case of non-invertible $\mathbf{A}$, one may utilize its Moore-Penrose inverse, often called the *pseudoinverse*, denoted as $\mathbf{A}^+$. It generalizes the notion of standard inverse of square matrices to those of arbitrary shape and rank. Consider the singular value decomposition (SVD) $\mathbf{A} = \mathbf{U}\mathbf{D}\mathbf{V}^*$ of $\mathbf{A}$ for matrices $\mathbf{U}, \mathbf{D}, \mathbf{V}$, where $\mathbf{U}$ and $\mathbf{V}$ are unitary, and $\mathbf{D}$ is diagonal with non-negative real entries. Then, its pseudoinverse is computed via $\mathbf{A}^+ = \mathbf{V}\mathbf{D}^+\mathbf{U}^*$, where $\mathbf{D}^+$ is obtained by replacing all of its non-zero singular values with their reciprocals.

**Range-nullspace decomposition.** A particularly useful property of the matrix pseudoinverse is its role in the well-known range-nullspace decomposition [Strang, 2022]. Specifically, any signal $\mathbf{x}$ can be decomposed as

$$\mathbf{x} = \underbrace{\mathbf{A}^+\mathbf{A}\mathbf{x}}_{range\ space} + \underbrace{(\mathbf{I} - \mathbf{A}^+\mathbf{A})\mathbf{x}}_{null\ space}, \tag{5}$$

where the first term lies in the range (image) of $\mathbf{A}$, and the second in its null space (kernel), *i.e.*, it is annihilated by $\mathbf{A}$. This decomposition is particularly relevant in inverse problems: applying $\mathbf{A}$ to $\mathbf{x}$ eliminates the null space component, while preserving the range component. Consequently, solving an inverse problem can be viewed as denoising in the range space and synthesizing missing information in the null space.

**Pseudoinverse reconstruction.** A matrix pseudoinverse can also be used to compute the *pseudoinverse reconstruction* (PR) $\hat{\mathbf{x}}$ of $\mathbf{x}$ via $\hat{\mathbf{x}} = \mathbf{A}^+\mathbf{y}$. Crucially, this reconstruction always lies in $\mathbb{R}^d$, regardless of the shape of $\mathbf{y}$. This property addresses a common limitation of prior bridge methods, *i.e.*, their default operation in fixed-dimensional spaces. When $m \neq d$, such methods require additional effort to remain well-defined. For more challenging inverse problems, such as those in medical imaging, PR provides a principled mechanism for resolving this issue.

From a theoretical perspective, assuming access to $\mathbf{A}^+$ also simplifies the relationship between $\mathbf{x}$ and $\mathbf{y}$ [Strang, 2022]. When $m < d$, the PR is the unique minimizer of $\|\mathbf{A}\mathbf{x}^* - \mathbf{y}\|_2$ over all $\mathbf{x}^* \in \mathbb{R}^d$. When $m > d$, it yields the minimum $\ell_2$-norm solution among those satisfying $\mathbf{A}\mathbf{x}^* = \mathbf{y}$, *i.e.*, $\min_{\mathbf{x}^*} \|\mathbf{x}^*\|_2$ subject to $\mathbf{A}\mathbf{x}^* = \mathbf{y}$. These optimality properties further motivate the use of PR when $\mathbf{A}$ is known and its pseudoinverse is tractable.

**Problem-specific diffusion bridges.** Many unsupervised approaches, assuming access to a specific noisy linear system (eq. (4)), offer tailored solutions that improve performance under additional assumptions [Song et al., 2023a, Wang et al., 2023, Chung et al., 2023a, Garber and Tirer, 2024]. In contrast, state-of-the-art (SOTA) diffusion bridges [Liu et al., 2023, Luo et al., 2023a, Yue et al., 2024, Zhou et al., 2024] make no assumptions about the underlying system during training, focusing on general mappings between arbitrary distributions. This can be suboptimal when the system is (even approximately) known. For instance, in image inpainting, these methods do not distinguish between the known range (unmasked) and the missing null part (masked), resulting in redundant noise in the range space, which is expected to be noiseless. In more complex scenarios, such redundancy could reduce efficiency, while ignoring system-specific information may hinder generalization. Thus, developing a supervised diffusion bridge tailored to a specific system remains an important yet unresolved challenge.

# 4 Method

In this section, we seek to construct a bridge diffusion process that directly incorporates the information about the assumed measurement system. By that, we understand a process which i) maps the PRs to the respective signal samples, ii) synthesizes the missing information directly in the null space and iii) optionally denoises the range space part. We begin by noticing a specific connection between how intermediate steps of a general linear diffusion process are obtained and how noisy linear measurement systems map clean samples to observations. As our key contribution, we propose a specific design of matrix-valued SDEs, which fulfill the initially desired properties.

**Connecting systems to SDEs.** Following eq. (4), it is clear that $\mathbf{y} \mid \mathbf{x} \sim \mathcal{N}(\mathbf{A}\mathbf{x}, \boldsymbol{\Sigma})$, *i.e.*, for a given signal $\mathbf{x}$, a noisy linear system applies a linear transformation $\mathbf{A}$ and adds (possibly correlated) Gaussian noise to sample a measurement. On the other hand, given a clean sample $\mathbf{x}_0$, eq. (1) transforms it to $\mathbf{x}_t$ by applying a *cascade* of such noisy linear systems, which is a noisy linear system itself. That is, $\mathbf{x}_t \mid \mathbf{x}_0 \sim \mathcal{N}(\mathbf{H}_t\mathbf{x}_0, \boldsymbol{\Sigma}_t)$ for some time-dependent matrices $\mathbf{H}_t, \boldsymbol{\Sigma}_t$. Hence, one may equivalently arrive at $\mathbf{y} \mid \mathbf{x}$ (or some linear transformation of $\mathbf{y}$) through an application of a series of noisy linear systems to $\mathbf{x}$ instead of just a single one. In the limit, this would imply the existence of a general linear SDE that performs such mapping.

The above considerations suggest that the measurement system could be directly *embedded* into $\mathbf{H}_t$ and $\boldsymbol{\Sigma}_t$ to arrive at such SDE. While the relationship between these and the drift and diffusion

coefficients of the corresponding stochastic process is not immediately clear, a recent theoretical result derived by Tivnan et al. [2025] provides a convenient way of mapping from one to the other under mild assumptions. We restate it below for clarity.

**Theorem 1** *[Tivnan et al., 2025] Assume that $\mathbf{x}_t \mid \mathbf{x}_0$ evolves according to the linear forward process from eq. (1). Then, $\mathbf{x}_t \mid \mathbf{x}_0 \sim \mathcal{N}(\mathbf{H}_t\mathbf{x}_0, \mathbf{\Sigma}_t)$, where*

$$\mathbf{H}_t = \exp\left(\mathbf{\Omega}_t(\mathbf{F}_t)\right) \approx \exp\left(\int_0^t \mathbf{F}_s ds\right), \mathbf{\Sigma}_t = \mathbf{H}_t\left(\int_0^t \mathbf{H}_s^{-1}\mathbf{G}_s\mathbf{G}_s^\top \mathbf{H}_s^{-1^\top} ds\right)\mathbf{H}_t^\top, \quad (6)$$

*where $\mathbf{\Omega}_t$ is the Magnus expansion and the approximation becomes equality if, for all $s, s' \in [0, t]$, $[\mathbf{F}_s, \mathbf{F}_{s'}] = 0$, i.e., $\mathbf{F}_s$ and $\mathbf{F}_{s'}$ commute. Conversely, given $\mathbf{H}_t$ and $\mathbf{\Sigma}_t$, the drift and diffusion coefficients can be obtained via:*

$$\mathbf{F}_t = \frac{d\mathbf{H}_t}{dt}\mathbf{H}_t^{-1}, \mathbf{G}_t\mathbf{G}_t^\top = \frac{d\mathbf{\Sigma}_t}{dt} - \mathbf{F}_t\mathbf{\Sigma}_t - \mathbf{\Sigma}_t\mathbf{F}_t^\top. \quad (7)$$

Therefore, theorem 1 allows one to obtain the linear diffusion process governing the evolution of $\mathbf{x}_t \mid \mathbf{x}_0$ defined through $\mathbf{H}_t$ and $\mathbf{\Sigma}_t$, which in turn makes training and sampling from continuous-time models possible.

**Embedding the measurement system.** Equipped with the necessary tools, we propose to embed the measurement system into the linear diffusion process by using

$$\mathbf{H}_t = \mathbf{A}^+\mathbf{A} + \alpha_t(\mathbf{I} - \mathbf{A}^+\mathbf{A}), \quad (8a)$$

$$\mathbf{\Sigma}_t = \gamma_t\mathbf{A}^+\mathbf{\Sigma}\mathbf{A}^{+\top} + \beta_t(\mathbf{I} - \mathbf{A}^+\mathbf{A}). \quad (8b)$$

for $\alpha_t, \beta_t, \gamma_t$ being the null space drift, null space diffusion and range space diffusion coefficients respectively. We refer to the process resulting from eqs. (8a) and (8b) as the *System-embedded Diffusion Bridge Model* (SDB).

To better understand the rationale behind this specific design, it is worth considering the range and null space components of the resulting $\mathbf{x}_t$ separately. Specifically,

$$\mathbf{A}^+\mathbf{A}\mathbf{x}_t = \mathbf{A}^+(\mathbf{A}\mathbf{x}_0 + \gamma_t\mathbf{\Sigma}^{\frac{1}{2}}\boldsymbol{\epsilon}) = \mathbf{A}^+\mathbf{A}\mathbf{x}_0 + \sqrt{\gamma_t}\mathbf{A}^+\mathbf{\Sigma}^{\frac{1}{2}}\boldsymbol{\epsilon}, \quad (9a)$$

$$(\mathbf{I} - \mathbf{A}^+\mathbf{A})\mathbf{x}_t = \alpha_t(\mathbf{I} - \mathbf{A}^+\mathbf{A})\mathbf{x}_0 + \sqrt{\beta_t}(\mathbf{I} - \mathbf{A}^+\mathbf{A})\boldsymbol{\epsilon}', \quad (9b)$$

where $\boldsymbol{\epsilon} \sim \mathcal{N}(\mathbf{0}_m, \mathbf{\Sigma}_{m \times m}), \boldsymbol{\epsilon}' \sim \mathcal{N}(\mathbf{0}_d, \mathbf{\Sigma}_{d \times d})$. The range space part contains the original signal $\mathbf{A}^+\mathbf{A}\mathbf{x}_0$ and the stochastic component $\gamma_t\mathbf{A}^+\mathbf{\Sigma}^{\frac{1}{2}}\boldsymbol{\epsilon}$, which directly models the range space noise. If $\mathbf{\Sigma} = \mathbf{0}$, the true signal is fully recovered at every timestep $t$. The null space part is a mixture of the null component of the true signal $\alpha_t(\mathbf{I} - \mathbf{A}^+\mathbf{A})\mathbf{x}_0$ and null-space-projected Gaussian noise $\beta_t(\mathbf{I} - \mathbf{A}^+\mathbf{A})\boldsymbol{\epsilon}'$. Notably, SDB links two stochastic processes that evolve simultaneously within the range and null spaces, each modeled with independent noise variables, $\boldsymbol{\epsilon}$ and $\boldsymbol{\epsilon}'$. Moreover, it is evident that proper choices for $\alpha_t, \beta_t$ and $\gamma_t$ lead to a mapping between the PRs at $t = 1$ to their respective clean samples at $t = 0$.

**SDE perspective.** Applying theorem 1 to the coefficients from eq. (8) leads to the following drift and diffusion coefficients for SDB:

$$\mathbf{F}_t = \frac{\mathrm{d}}{\mathrm{d}t}\log\alpha_t(\mathbf{I} - \mathbf{A}^+\mathbf{A}), \quad (10a)$$

$$\mathbf{G}_t\mathbf{G}_t^\top = \frac{\mathrm{d}\gamma_t}{\mathrm{d}t}\mathbf{A}^+\mathbf{\Sigma}\mathbf{A}^{+\top} + \left(\frac{\mathrm{d}\beta_t}{\mathrm{d}t} - 2\beta_t\frac{\mathrm{d}}{\mathrm{d}t}\log\alpha_t\right)(\mathbf{I} - \mathbf{A}^+\mathbf{A}). \quad (10b)$$

With this formulation at hand, we now propose a specific setting of the scalar functions $\alpha_t$ and $\beta_t$, which extend the result of Liu et al. [2023] and make direct connection of SDB to the optimal transport (OT) plan [Mikami, 2004].

**Theorem 2** *For the linear SDE defined in eq. (1) with $\mathbf{F}_t$ and $\mathbf{G}_t$ given by eqs. (10a) and (10b) respectively, let $\alpha_t = \frac{\bar{\sigma}_t^2}{\bar{\sigma}_t^2 + \sigma_t^2}, \beta_t = \frac{\sigma_t^2\bar{\sigma}_t^2}{\bar{\sigma}_t^2 + \sigma_t^2}$ for $\sigma_t^2 = \int_0^t g^2(\tau)d\tau, \bar{\sigma}_t^2 = \int_t^1 g^2(\tau)d\tau$ for some non-negative function $g(t)$. Assuming that $g(t) \to 0$ uniformly for all $t$, the null space part of this SDE reduces to an OT-ODE:*

$$d(\mathbf{I} - \mathbf{A}^+\mathbf{A})\mathbf{x}_t = \mathbf{v}_t(\mathbf{x}_t \mid \mathbf{x}_0)dt, \quad (11)$$

*where $\mathbf{v}_t(\mathbf{x}_t \mid \mathbf{x}_0) = \left(\lim_{g(t) \to 0}\frac{g^2(t)}{\sigma_t^2}\right)(\mathbf{I} - \mathbf{A}^+\mathbf{A})(\mathbf{x}_t - \mathbf{x}_0)$.*

We include the proof in the Appendix. To align the dynamics between the range and null space, we also use $\gamma_t = \frac{\sigma_t^2}{\bar{\sigma}_t^2 + \sigma_t^2}$. In practice, we do not set $g(t) = 0$, but rather keep it at sufficiently low values. Because of the relationship to the SB problem, we term this variant as SDB (SB).

**Novel processes.** To showcase the versatility of our framework, we introduce two additional variants of SDB that reinterpret the VP and VE diffusion processes of Song et al. [2021b]:

**SDB (VP):** $\alpha_t = 1 - t, \beta_t = \sqrt{1 - \alpha_t}, \gamma_t = \beta_t$,

**SDB (VE):** $\alpha_t = 1, \beta_t = \sigma_{max}\sqrt{t}, \gamma_t = \sqrt{t}$, where $\sigma_{max} \gg 1$.

In both cases, the original VP or VE process is applied in the null space, while the range space coefficient $\gamma_t$ is related to $\beta_t$ to simplify the dynamics. Unlike methods such as DDBM, which symmetrize the variance schedule around $t = 0.5$, these variants explicitly control how the original signal is erased in the null space. There, SDB (VP) performs a convex interpolation between the clean input and Gaussian noise, while SDB (VE) converges to an isotropic Gaussian $\mathcal{N}(\mathbf{0}, \sigma_{\max}^2\mathbf{I})$ as $t \to 1$. As such, both more closely resemble approaches like IR-SDE that retain non-singular marginals at the endpoint. For simplicity, we parameterize SDB (VP) and SDB (VE) processes with linear scheduling, leaving more sophisticated designs as future work.

**Principled posterior sampling.** In the asymptotic limit of infinite data and model capacity, it is natural to ask whether SDB constitutes an exact probabilistic model of the underlying inverse problem—that is, whether it produces *principled posterior samples* from $p(\mathbf{x}|\mathbf{y})$. The following result provides a positive answer under mild conditions on the forward process coefficients, with the proof deferred to the Appendix.

**Theorem 3** *Let the forward measurement model be $p(\mathbf{y}|\mathbf{x}) = \mathcal{N}(\mathbf{Ax}, \mathbf{\Sigma})$. Then, under the SDB dynamics defined by eq. (1), eq. (7), and eq. (8), with time-dependent scalar coefficients $\alpha_t$, $\beta_t$, and $\gamma_t$ satisfying*

$$\lim_{t \to 1} \gamma_t = 1, \quad \lim_{t \to 1} \frac{\alpha_t^2}{\beta_t} = 0,$$

*the corresponding reverse-time SDE generates asymptotically exact samples from the Bayesian posterior distribution $p(\mathbf{x}|\mathbf{y})$.*

Table 1: Comparison of prior SOTA diffusion bridge methods with SDB. Subsequent rows denote the drift ($\mathbf{F}_t$) and diffusion ($\mathbf{G}_t$) coefficients of the forward process, while $\boldsymbol{\mu}_t$, $\mathbf{\Sigma}_t$ are its respective mean and covariance at timestep $t$. Each column follows the original notation of each paper.

| Method | I2SB | IR-SDE | GOUB | DDBM (VP) | SDB (SB) |
|---|---|---|---|---|---|
| $\mathbf{F}_t$ | $\frac{\beta_t}{\bar{\sigma}_t^2 + \sigma_t^2}(\mathbf{x}_1 - \mathbf{x}_0) - \beta_t \frac{\bar{\sigma}_t^4 - \sigma_t^4}{\sigma_t^2 \bar{\sigma}_2^2}(\mathbf{x}_t - \boldsymbol{\mu}_t)$ | $\theta_t(\mathbf{x}_1 - \mathbf{x}_t)$ | $(\theta_t + g^2(t)\frac{e^{-2\bar{\theta}_{t:1}}}{\sigma_{t:1}^2})(\mathbf{x}_1 - \mathbf{x}_t)$ | $(\frac{\mathrm{d}}{\mathrm{d}t}\log \alpha_t)\mathbf{x}_t + g^2(t)\frac{\left(\frac{\alpha_t}{\alpha_1}\mathbf{x}_1 - \mathbf{x}_t\right)}{\sigma_t^2\left(\frac{\mathrm{SNR}_t}{\mathrm{SNR}_1} - 1\right)}$ | $\frac{\mathrm{d}}{\mathrm{d}t}\log\alpha_t(\mathbf{I} - \mathbf{A}^+\mathbf{A})$ |
| $\mathbf{G}_t\mathbf{G}_t^\top$ | $\beta_t\mathbf{I}$ | $\sigma^2(t)\mathbf{I}$ | $g^2(t)\mathbf{I}$ | $g^2(t)\mathbf{I}$ | $\frac{\mathrm{d}\gamma_t}{\mathrm{d}t}\mathbf{A}^+\mathbf{\Sigma}\mathbf{A}^{+\top} + \left(\frac{\mathrm{d}\beta_t}{\mathrm{d}t} - 2\beta_t\frac{\mathrm{d}}{\mathrm{d}t}\log\alpha_t\right)(\mathbf{I} - \mathbf{A}^+\mathbf{A})$ |
| $\boldsymbol{\mu}_t$ | $\frac{\bar{\sigma}_t^2}{\bar{\sigma}_t^2 + \sigma_t^2}\mathbf{x}_0 + \frac{\sigma_t^2}{\sigma_t^2 + \bar{\sigma}_t^2}\mathbf{x}_1$ | $\mathbf{x}_1 + (\mathbf{x}_0 - \mathbf{x}_1)e^{-\bar{\theta}_{0:t}}$ | $e^{-\bar{\theta}_{0:t}}\frac{\sigma_{t:1}^2}{\sigma_{0:1}^2}\mathbf{x}_0 + [(1 - e^{-\bar{\theta}_{0:t}})\frac{\sigma_{t:1}^2}{\sigma_{0:1}^2} + e^{-2\bar{\theta}_{t:1}}\frac{\sigma_{0:t}^2}{\sigma_{0:1}^2}]\mathbf{x}_1$ | $\frac{\mathrm{SNR}_1\alpha_t}{\mathrm{SNR}_t\alpha_1}\mathbf{x}_1 + \alpha_t(1 - \frac{\mathrm{SNR}_1}{\mathrm{SNR}_t})\mathbf{x}_0$ | $\mathbf{A}^+\mathbf{Ax}_0 + \alpha_t(\mathbf{I} - \mathbf{A}^+\mathbf{A})\mathbf{x}_0$ |
| $\mathbf{\Sigma}_t$ | $\frac{\sigma_t^2\bar{\sigma}_t^2}{\bar{\sigma}_t^2 + \sigma_t^2}\mathbf{I}$ | $\lambda^2(1 - e^{-2\bar{\theta}_t})\mathbf{I}$ | $\frac{\sigma_{0:t}^2\sigma_{t:1}^2}{\sigma_{0:1}^2}\mathbf{I}$ | $\sigma_t^2(1 - \frac{\mathrm{SNR}_1}{\mathrm{SNR}_t})\mathbf{I}$ | $\gamma_t\mathbf{A}^+\mathbf{\Sigma}\mathbf{A}^{+\top} + \beta_t(\mathbf{I} - \mathbf{A}^+\mathbf{A})$ |
| Markovian | ✗ | ✗ | ✗ | ✗ | ✓ |
| Hyperparameters | $\beta_t, \sigma_t^2 = \int_0^t g^2(\tau)d\tau, \bar{\sigma}_t^2 = \int_t^1 g^2(\tau)d\tau$ | $\theta_t = \frac{\sigma_t^2}{\lambda^2}, \bar{\theta}_t = \int_0^t \theta_\tau d\tau$ | $\theta_t = \frac{g^2(t)}{2\lambda^2}, \bar{\theta}_{s:t} = \int_s^t \theta_\tau d\tau, \bar{\sigma}_{s:t}^2 = \frac{g^2(t)}{2\theta_t}(1 - e^{-2\bar{\theta}_{s:t}})$ | $\mathrm{SNR}_t = \frac{\alpha_t^2}{\sigma_t^2}, \alpha_t = \exp\left(-\frac{1}{2}\int_0^t g^2(\tau)d\tau\right), \sigma_t^2 = 1 - \alpha_t^2$ | $\alpha_t, \beta_t, \gamma_t$ follow from theorem 2 |

---

**Algorithm 1** SDB Training

---

**Require:** $p(\mathbf{x}_0), p(t), \mathbf{A}, \mathbf{A}^+, \mathbf{\Sigma}^{1/2}, \alpha_t, \beta_t, \gamma_t, \boldsymbol{D_\theta}$
1: **for** each iteration **do**
2:      $\mathbf{x} \sim p(\mathbf{x}_0)$                                           $\triangleright$ sample clean data $\mathbf{x}$
3:      $t \sim p(t)$                                            $\triangleright$ sample diffusion timestep $t$
4:      $\boldsymbol{\epsilon} \sim \mathcal{N}(\mathbf{0}, \mathbf{I}) \in \mathbb{R}^m$                       $\triangleright$ sample range-space Gaussian noise $\boldsymbol{\epsilon}$
5:      $\boldsymbol{\epsilon}' \sim \mathcal{N}(\mathbf{0}, \mathbf{I}) \in \mathbb{R}^d$                       $\triangleright$ sample null-space Gaussian noise $\boldsymbol{\epsilon}'$
6:      $\mathbf{x}_t \leftarrow [\mathbf{A}^+\mathbf{A} + \alpha_t(\mathbf{I} - \mathbf{A}^+\mathbf{A})]\mathbf{x} + \gamma_t^{\frac{1}{2}}\mathbf{A}^+\mathbf{\Sigma}^{\frac{1}{2}}\boldsymbol{\epsilon} + \beta_t^{\frac{1}{2}}(\mathbf{I} - \mathbf{A}^+\mathbf{A})\boldsymbol{\epsilon}'$    $\triangleright$ forward step
7:      $L_\theta \leftarrow \|\boldsymbol{D_\theta}(\mathbf{x}_t, t) - \mathbf{x}\|_1$                    $\triangleright$ compute reconstruction loss for $\mathbf{x}$
8:      $\theta \leftarrow \text{optimizer}(\nabla_\theta L_\theta)$                    $\triangleright$ update network parameters $\theta$
9: **end for**
10: **return** $\theta$

---

**Algorithm 2** SDB Sampling (Euler-Maruyama)

---

**Require:** $N, \mathbf{A}, \mathbf{A}^+, \mathbf{\Sigma}^{1/2}, \alpha_t, \beta_t, \gamma_t, \mathbf{H}_t, \mathbf{F}_t, \boldsymbol{D_\theta}, \mathbf{y}$
1:   $t \leftarrow 1$                                               $\triangleright$ initialize time
2:   $\Delta t \leftarrow 1/N$                                       $\triangleright$ set timestep
3:   $\boldsymbol{\epsilon}' \sim \mathcal{N}(\mathbf{0}, \mathbf{I}) \in \mathbb{R}^d$                       $\triangleright$ sample null-space noise
4:   $\hat{\mathbf{x}} \leftarrow \mathbf{A}^+\mathbf{y}$                          $\triangleright$ pseudoinverse reconstruction
5:   $\mathbf{x}_t \leftarrow \hat{\mathbf{x}} + \beta_t^{\frac{1}{2}}(\mathbf{I} - \mathbf{A}^+\mathbf{A})\boldsymbol{\epsilon}'$                 $\triangleright$ initializer
6:   **for** $i \in \{1, \ldots, N\}$ **do**
7:      $\boldsymbol{\epsilon} \sim \mathcal{N}(\mathbf{0}, \mathbf{I}) \in \mathbb{R}^m$                 $\triangleright$ sample range-space noise
8:      $\boldsymbol{\epsilon}' \sim \mathcal{N}(\mathbf{0}, \mathbf{I}) \in \mathbb{R}^d$                 $\triangleright$ sample null-space noise
       $\mathbf{x}_{t-\Delta t} \leftarrow \mathbf{x}_t + \Delta t\left[(f_t\mathbf{I} - 2\mathbf{F}_t)(\mathbf{H}_t\boldsymbol{D_\theta}(\mathbf{x}_t, t) - \mathbf{x}_t) - \mathbf{F}_t\mathbf{x}_t\right] +$
9:      $\Delta t^{\frac{1}{2}}\left[\left(\frac{\mathrm{d}\gamma_t}{\mathrm{d}t}\right)^{\frac{1}{2}}\mathbf{A}^+\mathbf{\Sigma}^{\frac{1}{2}}\boldsymbol{\epsilon} + \left(\frac{\mathrm{d}\beta_t}{\mathrm{d}t} - 2\beta_t\frac{\mathrm{d}}{\mathrm{d}t}\log\alpha_t\right)^{\frac{1}{2}}(\mathbf{I} - \mathbf{A}^+\mathbf{A})\boldsymbol{\epsilon}'\right]$   $\triangleright$ update
10:      $t \leftarrow t - \Delta t$                                $\triangleright$ decrement time
11:   **end for**
12:   **return** $\mathbf{x}_0$                                    $\triangleright$ final sample

---

## 5 Experiments

**Baselines.** We compare SDB with both supervised bridge methods and unsupervised plug-and-play diffusion-based baselines. For the former, we include I2SB [Liu et al., 2023], IR-SDE [Luo et al., 2023a], GOUB [Yue et al., 2024], and DDBM [Zhou et al., 2024]. For the latter, we pick DPS [Chung et al., 2023a], ΠGDM [Song et al., 2023a], and DDNM [Wang et al., 2023], all of which rely on the assumption of a noisy linear measurement model. Detailed descriptions of all baselines and other technical details are provided in the Appendix. Unlike these methods, which rely on scalar SDEs, SDB introduces a more general matrix-valued formulation that allows structured control over range and null space components. We summarize the resulting conceptual differences to baseline bridge methods in table 1, together with SDB's training and sampling procedures in algorithms 1 and 2. Following standard evaluation practice [Luo et al., 2023a, Yue et al., 2024], we report perceptual scores (FID [Heusel et al., 2017], LPIPS [Zhang et al., 2018]) and reconstruction metrics (PSNR, SSIM).

**Experimental design.** To isolate the contribution of SDB as a diffusion process, we standardize key implementation details across all methods. Specifically: (i) for both supervised and unsupervised approaches, we train score networks from scratch using the training hyperparameters and architecture of Luo et al. [2023a], with 256 training epochs for supervised methods and 512 for unsupervised ones; (ii) each supervised method learns a mapping between signal samples and their PRs, ensuring that SDB does not benefit from a favorable parameterization, particularly in settings where $m \neq d$; (iii) during evaluation, supervised methods use 100 discretization steps, while unsupervised methods use 200, along with a standard Euler–Maruyama solver and their optimal noise schedule (e.g., VP for DDBM). To ensure consistent evaluation, we reimplement all methods within a unified framework, allowing for the separation of algorithmic advancements from the proposed stochastic process.

Table 2: Quantitative comparison of SDB with the baselines across four inverse problems. **Bold** indicates best and underline second-best values for each metric.

| Method | Inpainting - CelebA-HQ | | | | Superresolution - DIV2K | | | | CT Reconstruction - RSNA | | | | MRI Reconstruction - Br35H | | | |
|---|---|---|---|---|---|---|---|---|---|---|---|---|---|---|---|---|
| | FID ↓ | LPIPS ↓ | PSNR ↑ | SSIM ↑ | FID ↓ | LPIPS ↓ | PSNR ↑ | SSIM ↑ | FID ↓ | LPIPS ↓ | PSNR ↑ | SSIM ↑ | FID ↓ | LPIPS ↓ | PSNR ↑ | SSIM ↑ |
| **Unsupervised** | | | | | | | | | | | | | | | | |
| DPS | 12.3 | 0.213 | 18.59 | 0.684 | 101.2 | 0.295 | 19.04 | 0.492 | 29.25 | 0.160 | 32.870 | 0.709 | 40.73 | 0.187 | 21.336 | 0.549 |
| IIGDM | 10.2 | 0.146 | 19.02 | 0.762 | 97.09 | 0.311 | 18.84 | 0.530 | 32.29 | 0.165 | 31.272 | 0.757 | 41.14 | 0.211 | 19.830 | 0.665 |
| DDNM | 10.7 | 0.112 | 18.90 | 0.791 | 99.21 | 0.302 | 17.13 | 0.513 | 30.41 | 0.238 | 27.226 | 0.783 | 42.23 | 0.196 | 18.122 | 0.636 |
| **Supervised** | | | | | | | | | | | | | | | | |
| I2SB | 5.56 | 0.047 | 27.41 | 0.889 | 83.73 | **0.176** | 25.21 | 0.686 | 24.81 | 0.107 | 41.886 | 0.924 | 31.54 | 0.065 | 28.750 | 0.849 |
| IR-SDE | 4.68 | 0.031 | 29.92 | 0.912 | 96.22 | 0.185 | 23.51 | 0.603 | 18.88 | 0.028 | 43.438 | 0.964 | 30.14 | 0.065 | 28.878 | 0.871 |
| GOUB | 4.69 | 0.031 | 29.89 | 0.912 | 98.89 | 0.178 | 24.39 | 0.649 | 19.90 | 0.024 | 43.878 | 0.967 | 30.63 | 0.058 | 28.590 | 0.863 |
| DDBM | 6.03 | 0.047 | 28.15 | 0.906 | 90.16 | 0.233 | 25.79 | 0.720 | 23.36 | 0.040 | 44.415 | 0.964 | 32.42 | 0.074 | 28.971 | 0.872 |
| **Ours** | | | | | | | | | | | | | | | | |
| SDB (VP) | 4.90 | **0.030** | 30.51 | 0.914 | 87.08 | 0.228 | 25.91 | **0.724** | 15.43 | 0.019 | 46.365 | 0.981 | 32.88 | 0.068 | 29.255 | 0.881 |
| SDB (VE) | 5.97 | 0.042 | **32.12** | **0.944** | 94.73 | 0.226 | 25.90 | 0.718 | **14.17** | 0.020 | 46.325 | 0.981 | 33.90 | 0.083 | 29.098 | 0.876 |
| SDB (SB) | **4.63** | 0.031 | 30.40 | 0.930 | **81.56** | 0.226 | **26.10** | 0.724 | 15.02 | **0.018** | **46.672** | **0.982** | **29.85** | **0.053** | **29.812** | **0.893** |

**Benchmarks.** We evaluate SDB on four inverse problems with varying measurement system complexities, using original images at a resolution of $256 \times 256$. Building on prior work [Luo et al., 2023a, Yue et al., 2024], we first consider **inpainting on CelebA-HQ** [Karras et al., 2018], where $\mathbf{A}$ is a masking operator and $\mathbf{\Sigma} = \mathbf{0}_{m \times m}$ (noiseless). The measurements consist of the masked original images ($m = d$), which are equivalent to the PRs ($\mathbf{A}^+ = \mathbf{I}$). Additionally, we follow previous studies by examining **superresolution on DIV2K** [Agustsson and Timofte, 2017, Timofte et al., 2017], where $\mathbf{A}$ represents a $4\times$ downsampling operator ($d = 4m$), implemented through average pooling. For this task, $\mathbf{A}^+$ reconstructs the images using nearest-neighbor interpolation. To evaluate more complex and practical inverse problems, we additionally propose two medical imaging tasks: **CT reconstruction on the RSNA Intracranial Hemorrhage dataset** [Anouk Stein et al., 2019] and **MRI reconstruction on the Br35H dataset** [Merlin, 2022], both using 2D axial brain scan slices.

For CT reconstruction, the sinogram $\mathbf{y}$ represents line integrals of the object's attenuation coefficient, with the system matrix $\mathbf{A}$ describing detector-specific integrations along X-ray trajectories. We implement $\mathbf{A}$ using its SVD decomposition, $\mathbf{A} = \mathbf{UDV}^*$. To make the problem more realistic and introduce domain-specific artifacts, we zero the singular values of $\mathbf{D}$ below a threshold $\tau = 3.2$ and use a noisy setting with scalar covariance $\mathbf{\Sigma} = \sigma_1^2 \mathbf{I}$, where $\sigma_1^2 = 0.0001$.

For MRI reconstruction, $\mathbf{y}$ consists of undersampled Fourier-domain measurements. The system matrix is modeled as $\mathbf{A} = \mathbf{M}\mathcal{F}$, where $\mathcal{F}$ is the Fourier transform and $\mathbf{M}$ is a masking matrix. We sample $\mathbf{M}$ such that $\lambda_1$ controls the percentage of the lowest frequencies to be kept (a deterministic operation), while $\lambda_2$ specifies the percentage of frequencies to sample from the remaining ones (a stochastic operation). The noisy setting is similarly modeled with a scalar covariance matrix $\mathbf{\Sigma} = \sigma_2^2 \mathbf{I}$, where $\sigma_2^2 = 5.0$. We pick $\lambda_1 = 16$ and $\lambda_2 = 30$ as defaults.

## 5.1 General evaluation

The results summarized in table 2 show that SDB (SB) outperforms all baseline methods across every metric, with the exception of LPIPS on DIV2K. Notably, SDB (SB) demonstrates a clear advantage over I2SB, which shares the most similar stochastic process, highlighting how SDB (SB) effectively leverages additional information from the measurement system. Overall, it also provides empirical justification for the null space OT plan mentioned in theorem 2. Moreover, SDB (SB) displays superior stability, as the performance rankings of the baseline bridge methods exhibit more variability across tasks. These advantages are also evident qualitatively (fig. 2).

SDB (VE) and SDB (VP) strike a different balance between perceptual and reconstruction metrics compared to SDB (SB). In inpainting, they often outperform all other methods, with a noticeable emphasis on reconstruction quality. This trend is even more pronounced in the superresolution and MRI reconstruction tasks. In CT reconstruction, their performance is nearly on par with SDB (SB). Although they occasionally underperform relative to baseline bridge methods, this is not unexpected given the nature of their stochastic process. Along with IR-SDE, these methods are unique in not treating $p(\mathbf{x}_1)$ as a sum of Dirac deltas, which means they start the reverse process from a noisy sample. This additional noise may make the problem more challenging, but when directly comparing SDB (VE) and SDB (VP) to IR-SDE, the performance improvements become evident.

Finally, under our unified setting, prior bridge methods demonstrate comparable performance with significantly reduced variability compared to what previous works report [Luo et al., 2023a, Yue

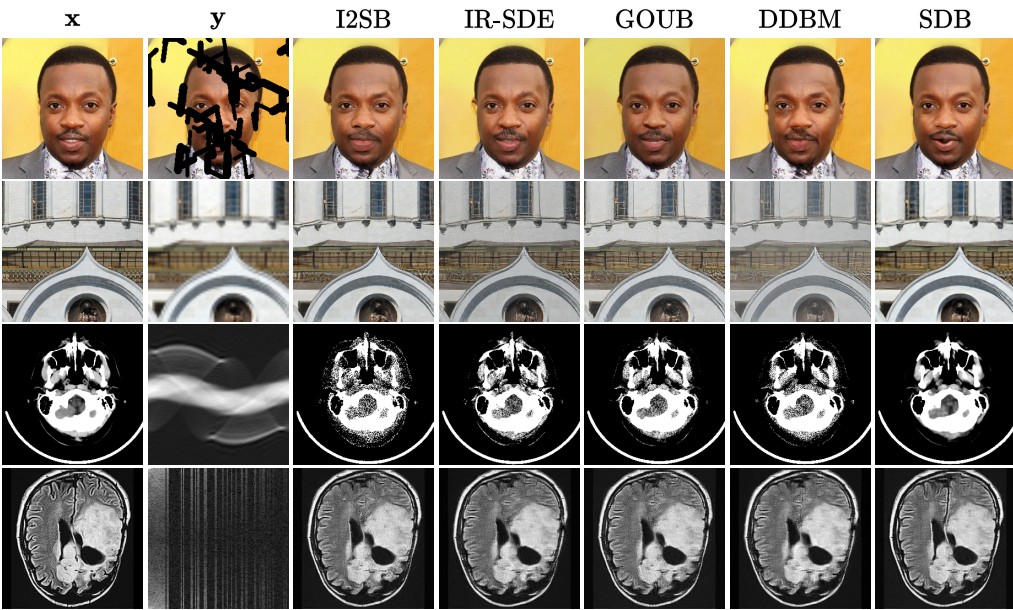

| **x** | **y** | I2SB | IR-SDE | GOUB | DDBM | SDB |
|---|---|---|---|---|---|---|

Figure 2: Qualitative comparison of SDB (SB) with the best-performing baselines (bridge methods). Rows depict the results for inpainting, superresolution, CT and MRI reconstruction respectively.

et al., 2024]. For instance, their PSNR on the MRI reconstruction task falls within the range of $[28.590, 28.971]$, meaning that performance gains on the order of $\approx 0.2$ can be considered significant in this context. Notably, the unsupervised baselines exhibit visibly lower performance compared to bridge methods. While this is expected due to the lack of training with paired data, we note that our setup allocates only twice the training budget to the unconditional models compared to the bridge methods, suggesting that their performance has not yet saturated. We consider improving these models as future work.

## 5.2 Evaluation under a misspecified model

In practice, while the general form of the measurement system in an inverse problem may be known, the specific parameter values often differ between training and deployment. This creates a generalization challenge, where the model must learn the underlying dynamics rather than relying on shortcuts. For example, in CT reconstruction, a well-trained model should perform stably near the original parameter $\tau$. A larger $\tau'$, which reduces low-frequency information, makes the task more difficult. However, a generalized model should maintain most of its performance as long as $\tau'$ remains reasonably close to $\tau$.

This issue is particularly critical for SDB, where the measurement system's parameters are embedded directly into the coefficients of its stochastic process. While results in section 5.1 indicate performance gains, deploying SDB in practice could be risky if it overfits to specific parameter values. Therefore, the following experiment tests whether SDB can generalize when evaluated under a misspecified model. For this evaluation, we focus on the best-performing methods from section 5.1, namely the baseline bridge methods and the SB variant of SDB.

**Improved generalization.** We begin with the MRI reconstruction task, where the system response matrix preserves a portion of the original signal by keeping $\lambda_1$ of the frequencies, starting from the lowest. This maintains the general structure of the true image, and as $\lambda_1$ is gradually decreased, more information is lost, making the task increasingly difficult. This scenario also mirrors real-world situations where a detector captures fewer measurements.

In what follows, we evaluate the checkpoints of bridge methods trained under a default setting $(\lambda_1 = 16, \lambda_2 = 30, \sigma_2^2 = 5.0)$ on measurements generated from a system with modified parameters.

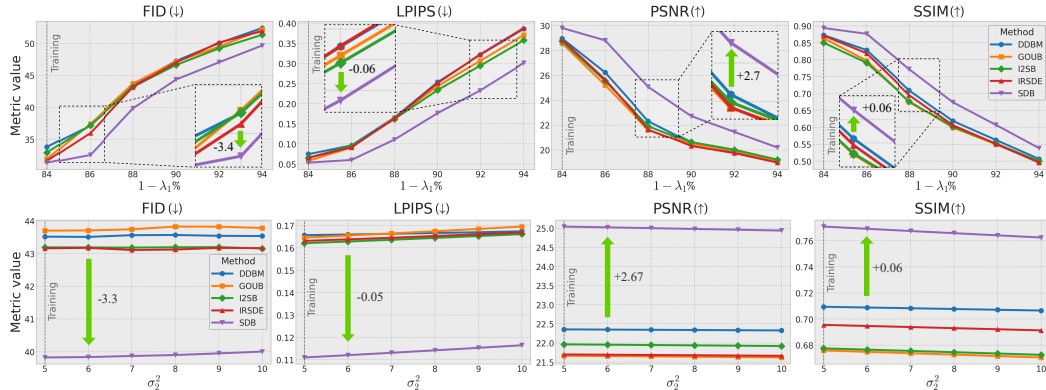

Figure 3: Quantitative comparison of SDB (SB) with other bridge methods in a misspecified MRI reconstruction setting. The **top** row evaluates bridges trained with $\lambda_1 = 16$, $\sigma_2^2 = 5$ on data generated from systems with decreasing $\lambda_1$. The **bottom** row evaluates performance on data with $\lambda_1 = 14$ and increasing $\sigma_2^2$. Perturbing the original system makes the problem harder in both cases.

Figure 3 (top) shows the performance of the methods as $\lambda_1$ is gradually decreased. Notably, the performance gap between SDB and the baseline methods widens as early as $\lambda_1 = 14$, and this advantage is sustained even at lower values. SDB continues to better utilize the available measurement information, indicating much greater robustness to reductions in $\lambda_1$.

To make the setting even more challenging, we set $\lambda_1 = 14$ and gradually increase the measurement noise variance $\sigma_2^2$ to up to twice its training-time value, simulating a more realistic scenario. Figure 3 (bottom) shows that, even under this perturbation, SDB maintains a significant performance advantage over the baseline methods. Unsurprisingly, all methods exhibit greater robustness to changes in $\sigma_2^2$, which is expected due to their denoising nature.

An analogous analysis for the CT reconstruction task is provided in the Appendix, yielding similar observations. Both experiments, closely aligned with the practical applications of inverse problem solvers, underscore the generalization capabilities of SDB across a broad range of system perturbations, positioning it as a promising solution for real-world scenarios.

**Extended evaluation.** Additional experimental results are presented in the Appendix. These include evaluations on more challenging misspecified models, analyses of the influence of ill-conditioned system operators, and detailed assessments of computational complexity and runtime. We also include a series of ablation studies. Finally, we outline a conceptual extension of SDB to nonlinear measurement systems and demonstrate its empirical advantages over baseline bridge methods in a small-scale study. Extending SDB to a broader class of problems remains an interesting direction for future work.

## 6 Discussion and limitations

SDB offers a principled framework for constructing measurement-system-specific diffusion bridges tailored to inverse problems under a linear Gaussian model. By explicitly incorporating information about the measurement system into the generative process, it achieves improved reconstruction quality and exhibits strong generalization under system perturbations. However, the method also comes with natural limitations. Notably, extending SDB to nonlinear measurement systems remains an open challenge. We provide only a preliminary, proof-of-concept treatment based on local linearization, which is limited to differentiable systems. In addition, our work adopts simple variance schedules for both the range and null space components; the interplay between these two processes warrants further theoretical and empirical investigation to identify optimal scheduling strategies. From a real-world application perspective, tasks such as CT and MRI reconstruction are typically performed in three dimensions—evaluating our method in such settings remains an open direction for future work. Despite these limitations, we view SDB as a valuable contribution to the growing literature on diffusion-based inverse problem solvers and a promising foundation for future research.

## Acknowledgments

This work was financially supported by the INFOSTRATEG-I/0022/2021-00 grant funded by the Polish National Centre for Research and Development (NCBiR), the SONATA BIS grant 2019/34/E/ST6/00052 funded by the Polish National Science Centre (NCN), and the NIH grant 5R01HL159183-03.

The computational resources for this work were provided by the Laboratory of Bioinformatics and Computational Genomics and the High Performance Computing Center of the Faculty of Mathematics and Information Science, Warsaw University of Technology. We also gratefully acknowledge Poland's High-performance Infrastructure PLGrid ACC Cyfronet AGH for providing computer facilities and support within computational grant no. PLG/2025/018330.

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

# Appendix for "System-Embedded Diffusion Bridge Models"

# A Theoretical results, proofs and derivations

## A.1 Derivation of drift and diffusion coefficients for SDB

We begin with deriving the drift and diffusion coefficients $\mathbf{F}_t, \mathbf{G}_t$ of SDB (eqs. (10a) and (10b)) by using theorem 1 with the mean and covariance matrices $\mathbf{H}_t, \boldsymbol{\Sigma}_t$ (eqs. (8a) and (8b)) of its forward process. Recall that $\mathbf{H}_t = \mathbf{A}^+\mathbf{A} + \alpha_t(\mathbf{I} - \mathbf{A}^+\mathbf{A}), \boldsymbol{\Sigma}_t = \gamma_t\mathbf{A}^+\boldsymbol{\Sigma}\mathbf{A}^{+\top} + \beta_t(\mathbf{I} - \mathbf{A}^+\mathbf{A})$. Following theorem 1, we obtain

$$\mathbf{A}^+\mathbf{A}\mathbf{F}_t = \frac{d}{dt}\mathbf{A}^+\mathbf{A}\log\mathbf{H}_t = \frac{d}{dt}\mathbf{A}^+\mathbf{A} = \mathbf{0}, \tag{12a}$$

$$(\mathbf{I} - \mathbf{A}^+\mathbf{A})\mathbf{F}_t = \frac{d}{dt}\log\alpha_t(\mathbf{I} - \mathbf{A}^+\mathbf{A}), \tag{12b}$$

$$\frac{d}{dt}\mathbf{A}^+\mathbf{A}\boldsymbol{\Sigma}_t = \frac{d\gamma_t}{dt}\mathbf{A}^+\boldsymbol{\Sigma}\mathbf{A}^{+\top}, \tag{12c}$$

$$\frac{d}{dt}(\mathbf{I} - \mathbf{A}^+\mathbf{A})\boldsymbol{\Sigma}_t = \frac{d\beta_t}{dt}(\mathbf{I} - \mathbf{A}^+\mathbf{A}). \tag{12d}$$

Moreover, note that $\mathbf{F}_t^\top = \mathbf{F}_t, \boldsymbol{\Sigma}_t^\top = \boldsymbol{\Sigma}_t$ due to the symmetry of the range and null space projections, and $\mathbf{F}_t\boldsymbol{\Sigma}_t = \boldsymbol{\Sigma}_t\mathbf{F}_t$ since $\mathbf{F}_t$ only affects the null space and $(\mathbf{I} - \mathbf{A}^+\mathbf{A})^2 = (\mathbf{I} - \mathbf{A}^+\mathbf{A})$ (idempotent). Following theorem 1,

$$\mathbf{F}_t = \frac{d}{dt}\log\alpha_t(\mathbf{I} - \mathbf{A}^+\mathbf{A}), \tag{13a}$$

$$\mathbf{G}_t\mathbf{G}_t = \frac{d\boldsymbol{\Sigma}_t}{dt} - \mathbf{F}_t\boldsymbol{\Sigma}_t - \boldsymbol{\Sigma}_t\mathbf{F}_t^\top \tag{13b}$$

$$= \frac{d\boldsymbol{\Sigma}_t}{dt} - 2\mathbf{F}_t\boldsymbol{\Sigma}_t \tag{13c}$$

$$= \frac{d\gamma_t}{dt}\mathbf{A}^+\boldsymbol{\Sigma}\mathbf{A}^{+\top} + \left(\frac{d\beta_t}{dt} - 2\beta_t\frac{d}{dt}\log\alpha_t\right)(\mathbf{I} - \mathbf{A}^+\mathbf{A}). \tag{13d}$$

## A.2 Proof of theorem 2

We begin by recalling two propositions of Liu et al. [2023] using our notation, which will serve as the basis of the proof.

**Proposition 1** *(Analytic posterior given boundary pair) [Liu et al., 2023] The posterior of eq. (1) given some boundary pair $(\mathbf{x}_0, \mathbf{x}_1)$ admits an analytic form:*

$$p(\mathbf{x}_t \mid \mathbf{x}_0, \mathbf{x}_1) = \mathcal{N}(\boldsymbol{\mu}_t(\mathbf{x}_0, \mathbf{x}_1), \boldsymbol{\Sigma}_t), \tag{14}$$

*where $\boldsymbol{\mu}_t = \frac{\bar{\sigma}_t^2}{\bar{\sigma}_t^2 + \sigma_t^2}\mathbf{x}_0 + \frac{\sigma_t^2}{\bar{\sigma}_t^2 + \sigma_t^2}\mathbf{x}_1, \boldsymbol{\Sigma}_t = \frac{\sigma_t^2\bar{\sigma}_t^2}{\bar{\sigma}_t^2 + \sigma_t^2}\mathbf{I}$ and $\sigma_t^2 = \int_0^t g^2(\tau)d\tau, \bar{\sigma}_t^2 = \int_t^1 = g^2(\tau)d\tau$ with $\mathbf{G}_t = g(t)\mathbf{I}$.*

**Proposition 2** *(Optimal Transport ODE; OT-ODE) [Liu et al., 2023] When $g^2(t) \to 0$, the SDE between $(\mathbf{x}_0, \mathbf{x}_1)$ reduces to an ODE:*

$$d\mathbf{x}_t = \mathbf{v}_t(\mathbf{x}_t \mid \mathbf{x}_0)dt, \tag{15}$$

*where $\mathbf{v}_t(\mathbf{x}_t \mid \mathbf{x}_0) = \frac{g^2(t)}{\sigma_t^2}(\mathbf{x}_t - \mathbf{x}_0)$ whose solution is the posterior mean of eq. (14).*

Consider the null space part of $\mathbf{x}_t$ given by mean and covariance matrices from eqs. (8a) and (8b) with $\alpha_t = \frac{\bar{\sigma}_t^2}{\bar{\sigma}_t^2 + \sigma_t^2}, \beta_t = \frac{\sigma_t^2\bar{\sigma}_t^2}{\bar{\sigma}_t^2 + \sigma_t^2}$ for $\sigma_t^2 = \int_0^t g^2(\tau)d\tau, \bar{\sigma}_t^2 = \int_t^1 g^2(\tau)d\tau$, where $g(t)$ is the null space diffusion coefficient given by eq. (10b), i.e., $g(t) = \left(\frac{d\beta_t}{dt} - 2\beta_t\frac{d}{dt}\log\alpha_t\right)^{\frac{1}{2}}$:

$$(\mathbf{I} - \mathbf{A}^+\mathbf{A})\mathbf{x}_t = \frac{\bar{\sigma}_t^2}{\bar{\sigma}_t^2 + \sigma_t^2}(\mathbf{I} - \mathbf{A}^+\mathbf{A})\mathbf{x}_0 + \left(\frac{\sigma_t^2\bar{\sigma}_t^2}{\bar{\sigma}_t^2 + \sigma_t^2}\right)^{\frac{1}{2}}(\mathbf{I} - \mathbf{A}^+\mathbf{A})\boldsymbol{\epsilon} \tag{16}$$

for $\epsilon \sim \mathcal{N}(\mathbf{0}_d, \mathbf{I}_{d \times d})$. Note that the null space part of $\mathbf{x}_1$, being the PR of $\mathbf{x}_0$, is zeroed out, *i.e.*, $(\mathbf{I} - \mathbf{A}^+\mathbf{A})\mathbf{x}_1 = (\mathbf{I} - \mathbf{A}^+\mathbf{A})\mathbf{0}_{d \times d}$. This allows us to artificially rewrite $\mathbf{x}_t$ as

$$(\mathbf{I} - \mathbf{A}^+\mathbf{A})\mathbf{x}_t = (\mathbf{I} - \mathbf{A}^+\mathbf{A})\left(\frac{\bar{\sigma}_t^2}{\bar{\sigma}_t^2 + \sigma_t^2}\mathbf{x}_0 + \frac{\sigma_t^2}{\bar{\sigma}_t^2 + \sigma_t^2}\mathbf{0}_{d \times d}\right) + \left(\frac{\sigma_t^2\bar{\sigma}_t^2}{\bar{\sigma}_t^2 + \sigma_t^2}\right)^{\frac{1}{2}}(\mathbf{I} - \mathbf{A}^+\mathbf{A})\epsilon. \quad (17)$$

Equivalently, $(\mathbf{I} - \mathbf{A}^+\mathbf{A})\mathbf{x}_t \sim \mathcal{N}((\mathbf{I} - \mathbf{A}^+\mathbf{A})\left(\frac{\bar{\sigma}_t^2}{\bar{\sigma}_t^2 + \sigma_t^2}\mathbf{x}_0 + \frac{\sigma_t^2}{\bar{\sigma}_t^2 + \sigma_t^2}\mathbf{x}_1\right), \frac{\sigma_t^2\bar{\sigma}_t^2}{\bar{\sigma}_t^2 + \sigma_t^2}(\mathbf{I} - \mathbf{A}^+\mathbf{A}))$. Hence, the posterior mean of SDB (SB) takes the form stated in proposition 1 and proposition 2 can be applied directly to the null space part.

As a final remark, we note that, up to this point, the definition of $g(t)$ is interdependent with that of $\alpha_t$ and $\beta_t$. It is not immediately clear that defining $\alpha_t, \beta_t$ as in theorem 2 fulfills $g^2(t) = \frac{\mathrm{d}\beta_t}{\mathrm{d}t} - 2\beta_t\frac{\mathrm{d}}{\mathrm{d}t}\log\alpha_t$. We now show that this property is indeed satisfied.

Denote by $C = \sigma_t^2 + \bar{\sigma}_t^2$ and observe that $\beta_t = \sigma_t^2\alpha_t, \bar{\sigma}_t^2 = C - \sigma_t^2, \frac{\mathrm{d}\sigma_t^2}{\mathrm{d}t} = g^2(t)$. Then,

$$\frac{\mathrm{d}}{\mathrm{d}t}\log\alpha_t = \frac{\mathrm{d}}{\mathrm{d}t}\log\frac{\bar{\sigma}_t^2}{\bar{\sigma}_t^2 + \sigma_t^2} \tag{18a}$$

$$= \frac{\mathrm{d}}{\mathrm{d}t}\left(\log\bar{\sigma}_t^2 - \log C\right) \tag{18b}$$

$$= \frac{g^2(t)}{\bar{\sigma}_t^2}, \tag{18c}$$

$$\frac{\mathrm{d}\beta_t}{\mathrm{d}t} = \frac{\mathrm{d}}{\mathrm{d}t}(\frac{\sigma_t^2\bar{\sigma}_t^2}{\bar{\sigma}_t^2 + \sigma_t^2}) \tag{18d}$$

$$= \frac{\mathrm{d}}{\mathrm{d}t}\left(\frac{\sigma_t^2(C - \sigma_t^2)}{C}\right) \tag{18e}$$

$$= \frac{1}{C}\frac{\mathrm{d}}{\mathrm{d}t}\left(C\sigma_t^2 - \sigma_t^4\right) \tag{18f}$$

$$= g^2(t) - \frac{2}{C}\sigma_t^2 g^2(t) \tag{18g}$$

$$= g^2(t)(1 - \frac{2}{C}\sigma_t^2). \tag{18h}$$

By substituting these into the definition of $g^2(t)$, we obtain

$$\frac{\mathrm{d}\beta_t}{\mathrm{d}t} - 2\beta_t\frac{\mathrm{d}}{\mathrm{d}t}\log\alpha_t = \frac{\mathrm{d}}{\mathrm{d}t}(\sigma_t^2\alpha_t) - 2\sigma_t^2\alpha_t\frac{\mathrm{d}}{\mathrm{d}t}\log\alpha_t \tag{19a}$$

$$= \alpha_t\frac{\mathrm{d}\sigma_t^2}{\mathrm{d}t} + \sigma_t^2\frac{\mathrm{d}\alpha_t}{\mathrm{d}t} - 2\sigma_t^2\alpha_t\left(\frac{1}{\alpha_t}\frac{\mathrm{d}\alpha_t}{\mathrm{d}t}\right) \tag{19b}$$

$$= \alpha_t g^2(t) - \sigma_t^2\frac{\mathrm{d}\alpha_t}{\mathrm{d}t} \tag{19c}$$

$$= \left(1 - \frac{\sigma_t^2}{C}\right)g^2(t) + \frac{\sigma_t^2}{C}g^2(t) \tag{19d}$$

$$= g^2(t), \tag{19e}$$

which completes the proof.

$\square$

## A.3  Proof of theorem 3

Many diffusion bridge methods, such as I2SB, GOUB, or DDBM, rely on non-Markovian stochastic processes, where $\mathbf{x}_t$ depends on both $\mathbf{x}_0$ and $\mathbf{x}_1$. In contrast, SDB uses a Markovian forward diffusion process, for which $p(\mathbf{x}_t|\mathbf{x}_0)$ is well-defined. More specifically, SDB is a special case of score-based generative models, using the general formulation with matrix-valued drift and diffusion coefficients described in Appendix A of [Song et al., 2021b]. We show below that SDB is a principled posterior sampler of $p(\mathbf{x}|\mathbf{y})$ with a proof of theorem 3 based on a series of lemmas.

We start by considering the most general practical case with a positive semidefinite covariance matrix $\Sigma$ in the forward model $p(\mathbf{y}' \mid \mathbf{x}) = \mathcal{N}(\mathbf{A}'\mathbf{x}, \Sigma)$. When $\Sigma$ is positive definite, we can perform whitening by scaling the measurements and the system matrix as $\mathbf{y} = \Sigma^{-\frac{1}{2}}\mathbf{y}'$ and $\mathbf{A} = \Sigma^{-\frac{1}{2}}\mathbf{A}'$, yielding the equivalent isotropic model $p(\mathbf{y} \mid \mathbf{x}) = \mathcal{N}(\mathbf{A}\mathbf{x}, \mathbf{I})$. In the singular case (i.e., $\Sigma$ is positive semidefinite but not positive definite), $\Sigma^{-\frac{1}{2}}$ can be replaced with $(\Sigma^{+})^{\frac{1}{2}}$ to achieve whitening on the range space of $\Sigma$. From this point onward, we drop primes for notational simplicity and proceed in the whitened coordinates. Hence, without loss of generality, we assume an isotropic noise model, which is equivalent to an initial whitening of both the measurements and the system response matrix.

**Lemma 1** *The pseudoinverse reconstruction $\mathbf{A}^{+}\mathbf{y}$ is a sufficient statistic for estimation of $\mathbf{x}$ given $\mathbf{y}$. That is,*

$$p(\mathbf{x}|\mathbf{y}) = p(\mathbf{x}|\mathbf{A}^{+}\mathbf{y}).$$

This holds because (i) the range space component of the measurements can be exactly recovered by re-applying $\mathbf{A}$ to $\mathbf{A}^{+}\mathbf{y}$, and (ii) the null space component contains no useful information about $\mathbf{x}$.

**Lemma 2** *A sample from*

$$p(\mathbf{z}|\mathbf{y}) = \mathcal{N}(\mathbf{A}^{+}\mathbf{y}, \beta_1(\mathbf{I} - \mathbf{A}^{+}\mathbf{A}))$$

*is also a sufficient statistic for estimating $\mathbf{x}$ given $\mathbf{y}$, i.e.,*

$$p(\mathbf{x}|\mathbf{y}) = p(\mathbf{x}|\mathbf{z}).$$

The null-space component of the pseudoinverse reconstruction is zero; therefore, adding null-space noise preserves sufficiency.

**Lemma 3** *The SDB forward process at the final time step, $p(\mathbf{x}_1|\mathbf{x}_0 = \mathbf{x})$, is identically distributed to the SDB initializer $p(\mathbf{z}|\mathbf{x})$ from lemma 2, assuming*

$$\lim_{t \to 1} \gamma_t = 1, \quad \lim_{t \to 1} \frac{\alpha_t^2}{\beta_t} = 0.$$

Both are conditional Gaussian random vectors with mean $\mathbf{A}^{+}\mathbf{A}\mathbf{x}$ and covariance $\mathbf{A}^{+}\Sigma\mathbf{A}^{+\top} + \beta_1(\mathbf{I} - \mathbf{A}^{+}\mathbf{A})$.

**Lemma 4** *The trained SDB model is a principled sampler of $p(\mathbf{x}_0|\mathbf{x}_1)$.*

Since SDB is a subset of score-based generative models with a standard Markovian forward process, the reverse SDE with a learned score network asymptotically samples from $p(\mathbf{x}_0|\mathbf{x}_1)$ [Anderson, 1982], approaching the exact score with sufficient data and model capacity.

The procedure to generate principled posterior samples from $p(\mathbf{x}|\mathbf{y})$ is as follows:

1. Compute the pseudoinverse reconstruction $\mathbf{A}^{+}\mathbf{y}$.

2. Sample null-space noise:

$$\mathbf{z} \sim p(\mathbf{z}|\mathbf{y}) = \mathcal{N}(\mathbf{A}^{+}\mathbf{y}, \beta_1(\mathbf{I} - \mathbf{A}^{+}\mathbf{A})).$$

3. Sample the SDB reverse process:

$$\mathbf{x}_0 \sim p(\mathbf{x}_0|\mathbf{x}_1 = \mathbf{z}).$$

By lemmas 1 and 2, $\mathbf{z}$ is a sufficient statistic for $\mathbf{x}$. Lemma 3 ensures that initializing the reverse process at $\mathbf{x}_1 = \mathbf{z}$ is valid. Lemma 4 guarantees that the trained SDB model samples from $p(\mathbf{x}_0|\mathbf{x}_1)$. Combining these results, the procedure produces principled posterior samples from $p(\mathbf{x}|\mathbf{y})$.

$\square$

### A.4 Magnus expansion generality

The matrix integral formulation of the reverse-time SDB process holds under the condition that the drift matrices $\mathbf{F}_t$ commute for all $t \in [0, 1]$. In the proposed SDB formulation (parameterized as in eq. (10a)), this condition is always satisfied: the only time-dependent parameter is the scalar $\alpha_t$, so the singular vectors of $\mathbf{F}_t$ remain constant, ensuring commutativity at all times.

This property extends to all scalar diffusion processes, including VP, VE, Flow Matching [Lipman et al., 2023], Fourier Diffusion Models Tivnan et al. [2025], and Subspace Diffusion Models Jing et al. [2022]. While diffusion methods with non-commuting $F_t$ matrices are currently not known to us, theorem 1 remains valid in those cases through the Magnus expansion Kamm et al. [2021].

## B Extended related works

### B.1 Diffusion models for inverse problems

In recent years, deep neural networks have gained significant attention for solving various inverse problems from different perspectives [Ongie et al., 2020, Chen et al., 2022]. Solving eq. (4) can be interpreted as sampling from the posterior distribution $p(\mathbf{x} \mid \mathbf{y})$, and generative models that support conditional generation are naturally suited to this task. Diffusion models, in particular, have emerged as SOTA tools for this purpose, thanks to their flexible mathematical formulation and expressive data priors [Daras et al., 2024, Preechakul et al., 2022, Sobieski and Biecek, 2024]. The standard approach involves extending the diffusion process to sample from $p(\mathbf{x}_0 \mid \mathbf{y})$ at $t = 0$ by applying Bayes' Theorem. The conditional score function can be decomposed as $\nabla_{\mathbf{x}_t} \log p(\mathbf{x}_t \mid \mathbf{y}) = \nabla_{\mathbf{x}_t} \log p(\mathbf{x}_t) + \nabla_{\mathbf{x}_t} \log p(\mathbf{y} \mid \mathbf{x}_t)$, where the first term, $\nabla_{\mathbf{x}_t} \log p(\mathbf{x}_t)$, can be approximated using a pretrained score network $\mathbf{s}_{\boldsymbol{\theta}}(\mathbf{x}_t, t)$, and the second term models the relationship between $\mathbf{x}_t$ and the measurement $\mathbf{y}$ [Song et al., 2021b, Dhariwal and Nichol, 2021]. This framework has spurred considerable progress in the field, with numerous successful methods emerging [Kawar et al., 2022, Chung et al., 2022, 2023a, Song et al., 2023a, Mardani et al., 2024, Chung et al., 2024, Sobieski et al., 2025]. Since this approach does not require retraining the score network for each specific problem, we refer to these methods as *unsupervised*.

Several works leverage the measurement system structure when applying pretrained diffusion models to inverse problems. Wang et al. [2023] (Denoising Diffusion Null-Space Models, DDNM) restrict updates during generation to the null space component of $\mathbf{x}_t$, keeping the range part fixed. Song et al. [2023a] (Pseudoinverse-Guided Diffusion Models, $\Pi$GDM) approximate the likelihood score via a vector-Jacobian product, where the score network's Jacobian is computed with automatic differentiation, and the vector reflects the range-nullspace decomposition. Garber and Tirer [2024] (Denoising Diffusion Models with Iteratively Preconditioned Guidance, DDPG) propose a method for interpolating between pseudoinverse-based and least-squares-based conditioning.

### B.2 Diffusion and Schrödinger bridges.

Diffusion models, while effective for high-quality image synthesis, are limited by the simplicity of Gaussian priors for $p(\mathbf{x}_1)$. Recent developments in *bridge models* [Särkkä and Solin, 2019] generalize the diffusion process by allowing $p(\mathbf{x}_1)$ to be an arbitrary distribution. This is especially important in image restoration tasks, where paired samples $(\mathbf{x}_0, \mathbf{x}_1)$—clean and distorted images—are available. Bridge models aim to generate samples from the posterior $p(\mathbf{x}_0 \mid \mathbf{x}_1)$ by initializing with a sample from $p(\mathbf{x}_1)$ rather than Gaussian noise. While conditioning the standard score network on $\mathbf{x}_1$ is a possible approach to achieve posterior sampling, it is often suboptimal [Batzolis et al., 2021].

Several methods have formulated the diffusion process as a stochastic bridge. Heng et al. [2021] propose a simulation-based algorithm using a fixed starting and ending point with an approximation of the true score. Liu et al. [2022] extend this with Doob's h-transform [Doob, 1984] to bridge distributions. Simulation-free algorithms utilizing the h-transform are presented by Somnath et al. [2023] and Peluchetti [2022]. Delbracio and Milanfar [2023] construct a Brownian Bridge for direct restoration from $\mathbf{x}_1$. More recently, Zhou et al. [2024] introduce the DDBM framework, extending the VE and VP processes, while He et al. [2024] study DDBM within the consistency framework [Song et al., 2023b]. Zheng et al. [2025] link DDBM to Denoising Diffusion Implicit Models [Song et al., 2021a].

The Schrödinger Bridge (SB) problem [Schrödinger, 1932], which aligns distributions via constrained forward and reverse SDEs, has also been explored. De Bortoli et al. [2021] apply Iterative Proportional Fitting (IPF) to solve the SB problem. Liu et al. [2023] propose a tractable class of SBs, leading to a simulation-free algorithm (I2SB). Chung et al. [2023b] extend it with an additional guidance term. Shi et al. [2024] build on IPF and introduce Iterative Markovian Fitting for SB solutions.

In a related approach, Luo et al. [2023a] derive a scalar case of eq. (1) that incorporates the start and end points, termed the *mean-reverting* SDE (IR-SDE). Through specific parameterization, they show that its score function is analytically tractable, connecting to the Ornstein-Uhlenbeck (OU) process [Gillespie, 1996]. Yue et al. [2024] extend this work with a generalized OU process and Doob's h-transform (GOUB). Further extensions include Luo et al. [2023b], who apply IR-SDE in latent spaces, and Welker et al. [2022] and Richter et al. [2023], who adapt similar processes for speech tasks. Recently, Zhu et al. [2025] unify GOUB and DDBM within the framework of stochastic optimal control.

### B.3 Baselines

#### B.3.1 Unsupervised

We begin with describing the unsupervised baselines, which rely on solving the following scalar reverse equation:

$$\mathrm{d}\mathbf{x}_t = [\mathbf{f}(\mathbf{x}_t, t) - g^2(t)\nabla_{\mathbf{x}_t} \log p(\mathbf{x}_t \mid \mathbf{y})]\mathrm{d}t + g(t)\mathrm{d}\overline{\mathbf{w}}_t, \tag{20}$$

where $\mathbf{f}$ and $g$ represent the drift and diffusion coefficients respectively, while $\mathbf{y}$ is the conditioning variable representing measurements in the inverse problem context. Unsupervised diffusion-based methods rely on decomposing the score function with Bayes' Theorem through $\nabla_{\mathbf{x}_t} \log p(\mathbf{x}_t \mid \mathbf{y}) = \nabla_{\mathbf{x}_t} \log p(\mathbf{x}_t) + \nabla_{\mathbf{x}_t} \log p(\mathbf{y} \mid \mathbf{x}_t)$ and approximating $\nabla_{\mathbf{x}_t} \log p(\mathbf{x}_t)$ with a pretrained $\mathbf{s}_{\boldsymbol{\theta}}(\mathbf{x}_t, t)$, while proposing different approaches for $\nabla_{\mathbf{x}_t} \log p(\mathbf{y} \mid \mathbf{x}_t)$.

Diffusion Posterior Sampling (DPS, Chung et al. [2023a]) approximates $p(\mathbf{y} \mid \mathbf{x}_t)$ with $p(\mathbf{y} \mid \hat{\mathbf{x}}_0(\mathbf{x}_t))$, where $\hat{\mathbf{x}}_0(\mathbf{x}_t) = \mathbb{E}[\mathbf{x}_0 \mid \mathbf{x}_t] = \mathbf{x}_t + g^2(t)\nabla_{\mathbf{x}_t} \log p(\mathbf{x}_t)$ is the Tweedie's formula [Robbins, 1992], giving an approximation of the denoised image at timestep $t$. Similarly, $\nabla_{\mathbf{x}_t} \log p(\mathbf{x}_t)$ is also approximated with $\mathbf{s}_{\boldsymbol{\theta}}(\mathbf{x}_t, t)$ in this case.

Pseudoinverse-Guided Diffusion Model ($\Pi$GDM, Song et al. [2023a]) proposes to approximate the loglikelihood score with $(\mathbf{y} - \mathbf{A}\hat{\mathbf{x}}_0(\mathbf{x}_t))^\top (r_t^2 \mathbf{A}\mathbf{A}^\top + \sigma_{\mathbf{y}}^2 \mathbf{I})^{-1} \mathbf{A}\frac{\partial \hat{\mathbf{x}}_0(\mathbf{x}_t)}{\partial \mathbf{x}_t}$, where $\sigma_{\mathbf{y}}^2 \mathbf{I}$ is the measurement system covariance and $r_t^2$ is a time-dependent term that should depend on the data (Appendix A.3., Song et al. [2023a]). Using automatic differentiation, one can compute $\frac{\partial \hat{\mathbf{x}}_0(\mathbf{x}_t)}{\partial \mathbf{x}_t}$. By combining it with the other terms, the entire approximation can be efficiently computed as a vector-Jacobian product.

In their basic formulation, Denoising Diffusion Null-space Models (DDNM, Wang et al. [2023]) rely on approximating $p(\mathbf{x}_{t-1} \mid \mathbf{x}_t, \mathbf{y})$ using the following update rule:

$$\mathbf{x}_{t-1} = \frac{\sqrt{\bar{\alpha}_{t-1}}\beta_t}{1 - \bar{\alpha}_t}\hat{\mathbf{x}}_0(\mathbf{x}_t, \mathbf{y}) + \frac{\sqrt{\alpha_t}(1 - \bar{\alpha}_{t-1})}{1 - \bar{\alpha}_t}\mathbf{x}_t + \sigma_t \boldsymbol{\epsilon}, \tag{21}$$

where $\boldsymbol{\epsilon} \sim \mathcal{N}(\mathbf{0}, \mathbf{I})$ and $\alpha_t, \beta_t$ follow the notation from the original work. This rule utilizes the range-nullspace decomposition via $\hat{\mathbf{x}}_0(\mathbf{x}_t, \mathbf{y}) = \mathbf{A}^+\mathbf{y} + (\mathbf{I} - \mathbf{A}^+\mathbf{A})\hat{\mathbf{x}}_0(\mathbf{x}_t)$, which preserves the true range space component, while updating the null space part with Tweedie's estimate.

#### B.3.2 Supervised

We proceed with a description of baseline supervised bridge methods. All of these approaches assume a stochastic process conditioned on both the starting and ending point with the following forward and reverse equations:

$$\mathrm{d}\mathbf{x}_t = \mathbf{f}(\mathbf{x}_t, \mathbf{x}_T, t, T)\mathrm{d}t + g(t)\mathrm{d}\mathbf{w}_t, \tag{22a}$$
$$\mathrm{d}\mathbf{x}_t = \mathbf{f}'(\mathbf{x}_t, \mathbf{x}_T, t, T)\mathrm{d}t + g(t)\mathrm{d}\overline{\mathbf{w}}_t, \tag{22b}$$

where $\mathbf{f}, \mathbf{f}'$ represent general drift coefficients respectively for the forward and reverse equation, $g$ is the diffusion coefficient and $\mathbf{x}_T$ represents the endpoint for the initial $\mathbf{x}_0$.

Image-to-Image Schrodinger Bridges (I2SB, Liu et al. [2023]) formulate the distribution of $\mathbf{x}_t$, given starting and ending points $\mathbf{x}_0, \mathbf{x}_T$, as $p(\mathbf{x}_t \mid \mathbf{x}_0, \mathbf{x}_T) = \mathcal{N}(\frac{\bar{\sigma}_t^2}{\bar{\sigma}_t^2 + \sigma_t^2}\mathbf{x}_0 + \frac{\sigma_t^2}{\bar{\sigma}_t^2 + \sigma_t^2}\mathbf{x}_T, \frac{\sigma_t^2 \bar{\sigma}_t^2}{\bar{\sigma}_t^2 + \sigma_t^2}\mathbf{I})$, where $\sigma_t^2 = \int_0^t g^2(\tau)\mathrm{d}\tau, \bar{\sigma}_t^2 = \int_t^1 g^2(\tau)\mathrm{d}\tau$ and show its equivalence to DDPM posterior sampling [Ho et al., 2020].

Image Restoration SDE (IR-SDE, Luo et al. [2023a]) is based on formulating the forward and reverse equations as

$$\mathrm{d}\mathbf{x}_t = \theta_t(\mathbf{x}_T - \mathbf{x}_t)\mathrm{d}t + \sigma(t)\mathrm{d}\mathbf{w}_t, \tag{23a}$$

$$\mathrm{d}\mathbf{x}_t = [\theta_t(\mathbf{x}_T - \mathbf{x}_t) - \sigma^2(t)\nabla_{\mathbf{x}_t}\log p(\mathbf{x}_t)]\mathrm{d}t + \sigma(t)\mathrm{d}\overline{\mathbf{w}}_t, \tag{23b}$$

where $\theta_t = \frac{\sigma_t^2}{\lambda^2}, \bar{\theta}_t = \int_0^t \theta_\tau d\tau$ and $\lambda$ is a predefined constant. At $t = 1$, IR-SDE arrives at a Gaussian distribution (with non-zero covariance) centered at $\mathbf{x}_T$.

Generalized Ornstein-Uhlenbeck Bridges (GOUB, Yue et al. [2024]) show IR-SDE as a special case of their framework by incorporating the Doob's h-transform [Doob, 1984], which pulls the process towards the desired endpoint. Formally, it is defined as $\mathbf{h}(\mathbf{x}_t, t, \mathbf{x}_T, T) = \nabla_{\mathbf{x}_t}\log p(\mathbf{x}_T \mid \mathbf{x}_t)$ and incorporated into the forward and reverse equations:

$$\mathrm{d}\mathbf{x}_t = \left( (\theta_t + g^2(t)\frac{e^{-2\bar{\theta}_{t:T}}}{\bar{\sigma}_{t:T}^2})(\mathbf{x}_T - \mathbf{x}_t) \right) \mathrm{d}t + g(t)\mathrm{d}\mathbf{w}_t, \tag{24a}$$

$$\mathrm{d}\mathbf{x}_t = \left( (\theta_t + g^2(t)\frac{e^{-2\bar{\theta}_{t:T}}}{\bar{\sigma}_{t:T}^2})(\mathbf{x}_T - \mathbf{x}_t) - g^2(t)\nabla_{\mathbf{x}_t}\log p(\mathbf{x}_t \mid \mathbf{x}_T) \right) \mathrm{d}t + g(t)\mathrm{d}\overline{\mathbf{w}}_t, \tag{24b}$$

where $\theta_t = \frac{g^2(t)}{2\lambda^2}, \bar{\theta}_{s:t} = \int_s^t \theta_\tau d\tau, \bar{\sigma}_{s:t}^2 = \frac{g^2(t)}{2\theta_t}(1 - e^{-2\bar{\theta}_{s:t}})$ for a predefined constant $\lambda$.

Denoising Diffusion Bridge Model, (DDBM, Zhou et al. [2024]) also utilize the h-transform, but instead show how to adapt prior unconditional processes, which map images to Gaussian noise, to construct a bridge between arbitrary distributions given paired data:

$$\mathrm{d}\mathbf{x}_t = [\mathbf{f}(\mathbf{x}_t, t) + g^2(t)\mathbf{h}(\mathbf{x}_t, t, \mathbf{x}_T, T)]\mathrm{d}t + g(t)\mathrm{d}\mathbf{w}_t, \tag{25a}$$

$$\mathrm{d}\mathbf{x}_t = [\mathbf{f}(\mathbf{x}_t, t) - g^2(t)\left(\nabla_{\mathbf{x}_t}\log p(\mathbf{x}_t \mid \mathbf{x}_T) - \mathbf{h}(\mathbf{x}_t, t, \mathbf{x}_T, T)\right)]\mathrm{d}t + g(t)\mathrm{d}\overline{\mathbf{w}}_t, \tag{25b}$$

where $\mathbf{f}(\mathbf{x}_t, t), g(t)$ follow the original image-to-noise process.

## C  Extended methodology

### C.1  Network parameterization

In a scalar setting, various reparameterizations of the score function were shown to provide different trade-offs in the final performance [Ho et al., 2020]. For example, the network could instead predict the added noise directly, which is bijectively obtained from the score. As these reparameterizations are related through simple scalar functions, one can choose them freely without additional considerations. In the matrix-valued setting, however, the choice of a parameterization is more subtle. In this case, observe that $\nabla_{\mathbf{x}_t}\log p(\mathbf{x}_t) = -\boldsymbol{\Sigma}_t^{-1}(\mathbf{x}_t - \mathbf{H}_t\mathbb{E}[\mathbf{x}_0 \mid \mathbf{x}_t])$. Training a score-prediction model hence requires access to the inverse of $\boldsymbol{\Sigma}_t$, which may in general be costly to obtain. By properly choosing the form of $\gamma_t$ and $\beta_t$, one can show that $\mathbf{G}_t\mathbf{G}_t^\top \nabla_{\mathbf{x}_t}\log p(\mathbf{x}_t) = (f_t\mathbf{I} - 2\mathbf{F}_t)(\mathbf{H}_t\mathbb{E}[\mathbf{x}_0 \mid \mathbf{x}_t] - \mathbf{x}_t)$ for some function $f_t$, i.e., an $\mathbf{x}_0$-prediction model alleviates the need for computing $\boldsymbol{\Sigma}_t^{-1}$. Therefore, we treat this parameterization as the default one for SDB.

### C.2  Nonlinear measurement systems

While SDB is primarily formulated for linear operators $\mathbf{A}$ with a global range-nullspace decomposition, it can be extended to certain nonlinear inverse problems. For a nonlinear, differentiable system response $\mathbf{A}$, we can approximate it locally using a first-order Taylor expansion around some point $\mathbf{x}_0$:

$$\mathbf{A}(\mathbf{x}) \approx \mathbf{A}(\mathbf{x}_0) + \boldsymbol{J}_{\mathbf{A}}(\mathbf{x}_0)(\mathbf{x} - \mathbf{x}_0), \tag{26}$$

where $\boldsymbol{J}_{\mathbf{A}}(\mathbf{x}_0)$ is the Jacobian of $\mathbf{A}$ at $\mathbf{x}_0$. This linearization induces a local range-nullspace decomposition, enabling the use of SDB on nonlinear systems. The procedure is as follows:

1. For a measurement $\mathbf{y} = \mathbf{A}(\mathbf{x}) + \mathbf{\Sigma}^{1/2}\epsilon$, compute an approximate maximum likelihood estimate $\hat{\mathbf{x}}$ of the signal $\mathbf{x}$.

2. Linearize the operator at $\hat{\mathbf{x}}$ using eq. (26) and compute the Jacobian $\mathbf{J_A}(\hat{\mathbf{x}})$.

3. Treat $\mathbf{y} \approx \mathbf{J_A}(\hat{\mathbf{x}})\mathbf{x} + \mathbf{\Sigma}^{1/2}\epsilon$ as a linear Gaussian model and apply standard SDB with pseudoinverse and range/nullspace projections.

This approach leverages the Jacobian as a local linear approximation of the nonlinear operator, effectively adapting SDB to more general, differentiable systems.

## D Experimental setup

### D.1 Training hyperparameters

We follow the training procedure proposed by Luo et al. [2023a], using the ADAM optimizer [Kingma and Ba, 2015] with an initial learning rate of $1 \times 10^{-4}$, no weight decay, and $(\beta_1, \beta_2) = (0.9, 0.99)$. A multi-step learning rate scheduler is applied, halving the learning rate at the 36th, 60th, 72nd, and 90th epochs, as in the original work. All methods are trained using the $\ell_1$ loss function with a batch size of 8. To ensure fairness across methods, we evaluate each model every 16 epochs and report the performance of the best checkpoint, rather than relying solely on the final one.

### D.2 Computational requirements

All experiments were conducted on a cluster of NVIDIA A100 GPUs, with each method trained using a single GPU. The approximate training times for each task are as follows: 4 hours for super-resolution, 24 hours for MRI reconstruction, 48 hours for CT reconstruction, and 72 hours for inpainting.

## E Additional experimental results

### E.1 Evaluation under a misspecified model (CT reconstruction)

Figure 4 presents the results of an analogous analysis of performance of bridge methods under a misspecified model (here in CT reconstruction task), where the first part of the experiment considers the default setting of $\sigma_1^2$ with increasing $\tau$, while the second part shows the results for $\tau = 3.6$ and increasing $\sigma_1^2$. In a similar manner to section 5.2, all SDB variants achieve a clear advantage over other bridge methods when the system's parameters are perturbed.

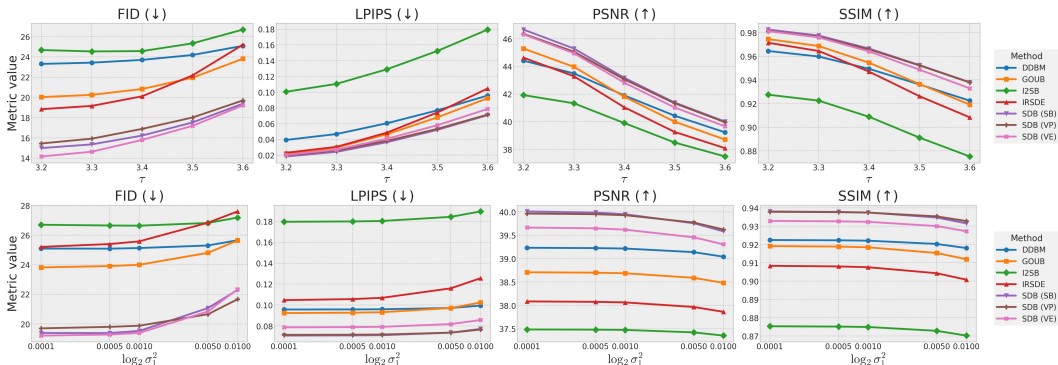

Figure 4: Quantitative comparison of SDB (SB) with other bridge methods in a misspecified CT reconstruction setting. The **top** row evaluates bridges trained with $\tau = 3.2$, $\sigma_1^2 = 0.0001$ on data generated from systems with increasing $\tau$. The **bottom** row evaluates performance on data with $\tau = 3.6$ and increasing $\sigma_1^2$. Perturbing the original system makes the problem harder in both cases.

## E.2 More challenging misspecified settings

We investigate more challenging and realistic misspecified scenarios for supervised inverse problems, including adversarial modifications of the forward operator and noise model. We evaluate SDB (SB) against supervised baselines across these settings.

### E.2.1 Misspecified operator

We consider two superresolution scenarios: a drastically enlarged downsampling kernel ($32 \times 32$) and JPEG compression at 30% quality. Both create inputs very different from training-time PRs. Table 3 reports results across both tasks, with the task indicated in the first column. SDB (SB) consistently achieves the best performance in 3 out of 4 metrics.

Table 3: Superresolution under more challenging operators (bold indicates best values).

| Method | Kernel $32 \times 32$ | | | | JPEG 30% | | | |
|---|---|---|---|---|---|---|---|---|
| | FID↓ | LPIPS↓ | PSNR↑ | SSIM↑ | FID↓ | LPIPS↓ | PSNR↑ | SSIM↑ |
| IRSDE | 130.47 | 0.684 | 17.85 | 0.381 | 99.43 | **0.283** | 24.81 | 0.651 |
| I2SB | 131.33 | **0.681** | 17.84 | 0.381 | 93.62 | 0.322 | 25.12 | 0.668 |
| DDBM | 131.55 | 0.700 | 17.83 | 0.380 | 92.39 | 0.373 | 25.19 | 0.685 |
| GOUB | 134.66 | 0.690 | 17.85 | 0.370 | 101.08 | 0.291 | 24.91 | 0.662 |
| SDB (SB) | **129.87** | 0.682 | **17.86** | **0.390** | **90.26** | 0.380 | **25.41** | **0.692** |

### E.2.2 Misspecified noise model

We also consider a misspecified noise model, replacing the Gaussian noise with a Poisson distribution to simulate photon-counting physics. For a given signal sample $\mathbf{x}$, measurements are sampled as $\mathbf{y} \sim \mathrm{Poisson}(\mathbf{I}_0 \cdot \exp(-\mathbf{A}\mathbf{x}))$, where $\mathbf{I}_0$ is the incident photon intensity. We evaluate CT reconstruction, where Poisson noise closely reflects real-world X-ray physics. As shown in table 4, SDB outperforms all supervised baselines across different $\mathbf{I}_0$ values, even without explicitly modeling the Poisson distribution.

Table 4: CT reconstruction with Poisson noise. Bold indicates best values.

| Method | $\mathbf{I}_0 = 10000$ | | | | $\mathbf{I}_0 = 5000$ | | | | $\mathbf{I}_0 = 1000$ | | | |
|---|---|---|---|---|---|---|---|---|---|---|---|---|
| | FID↓ | LPIPS↓ | PSNR↑ | SSIM↑ | FID↓ | LPIPS↓ | PSNR↑ | SSIM↑ | FID↓ | LPIPS↓ | PSNR↑ | SSIM↑ |
| GOUB | 17.78 | 0.0230 | 44.85 | 0.9717 | 17.83 | 0.0240 | 44.59 | 0.9705 | 20.52 | 0.0491 | 42.79 | 0.9588 |
| IRSDE | 17.29 | 0.0256 | 44.17 | 0.9681 | 17.86 | 0.0299 | 43.88 | 0.9665 | 21.12 | 0.0678 | 41.84 | 0.9506 |
| DDBM | 20.69 | 0.0402 | 44.11 | 0.9622 | 20.57 | 0.0406 | 43.93 | 0.9613 | 21.75 | 0.0575 | 42.53 | 0.9525 |
| I2SB | 21.91 | 0.1014 | 41.68 | 0.9242 | 21.82 | 0.1018 | 41.54 | 0.9231 | 22.43 | 0.1170 | 40.44 | 0.9130 |
| SDB (SB) | **14.03** | **0.0193** | **46.05** | **0.9799** | **14.61** | **0.0203** | **45.70** | **0.9785** | **19.80** | **0.0461** | **43.44** | **0.9657** |

We further consider MRI reconstruction under Poisson noise as an adversarial setting, where no natural process is typically modeled with Poisson noise, making it highly out-of-distribution. Table 5 shows that SDB again outperforms other methods in 3 out of 4 metrics while almost matching the best SSIM.

### E.3 Motion deblurring with an ill-conditioned system

To assess the practicality of SDB in ill-conditioned systems, we consider a motion deblurring task on $128 \times 128$ flower images from the Flowers102 dataset [Nilsback and Zisserman, 2008]. The system matrix is modeled as a block Toeplitz matrix with Toeplitz blocks (BTTB), implemented via 2D convolutions in the pixel domain. For the pseudoinverse approximation, we apply zero-order Tikhonov regularization (Wiener filtering) in the frequency domain. Although this differs from the exact pseudoinverse, SDB matches the best performance in LPIPS and clearly surpasses all supervised baselines in the other three metrics, demonstrating robustness to ill-conditioned operators (table 6).

### E.4 Contrast recovery with a nonlinear system

In the following, we provide an initial study concerning the application of SDB to nonlinear systems.

Table 5: MRI reconstruction with Poisson noise (bold indicates best values).

| Method | $\mathbf{I}_0 = 20.0$ | | | | $\mathbf{I}_0 = 16.0$ | | | |
|--------|-------|---------|-------|-------|-------|---------|-------|-------|
| | FID↓ | LPIPS↓ | PSNR↑ | SSIM↑ | FID↓ | LPIPS↓ | PSNR↑ | SSIM↑ |
| DDBM | 40.2161 | 0.2647 | 21.9509 | **0.5572** | 39.0941 | 0.2360 | 22.6901 | **0.5919** |
| GOUB | 39.8257 | 0.2821 | 21.4963 | 0.5246 | 39.3272 | 0.2530 | 22.2050 | 0.5577 |
| IRSDE | 39.8516 | 0.2706 | 21.5772 | 0.5383 | 39.4095 | 0.2445 | 22.2666 | 0.5722 |
| I2SB | 40.0352 | 0.2767 | 21.4514 | 0.5219 | 39.0571 | 0.2492 | 22.1715 | 0.5546 |
| SDB (SB) | **39.7303** | **0.2513** | **22.0684** | 0.5569 | **38.9677** | **0.2245** | **22.8237** | 0.5891 |

| Method | $\mathbf{I}_0 = 12.0$ | | | | $\mathbf{I}_0 = 8.0$ | | | |
|--------|-------|---------|-------|-------|-------|---------|-------|-------|
| | FID↓ | LPIPS↓ | PSNR↑ | SSIM↑ | FID↓ | LPIPS↓ | PSNR↑ | SSIM↑ |
| DDBM | 38.2960 | 0.2036 | 23.6083 | **0.6345** | 36.6739 | 0.1653 | 24.7777 | **0.6873** |
| GOUB | 38.4142 | 0.2196 | 23.0794 | 0.5991 | 36.9538 | 0.1794 | 24.2220 | 0.6524 |
| IRSDE | 38.4208 | 0.2141 | 23.1378 | 0.6139 | 37.0698 | 0.1769 | 24.2971 | 0.6680 |
| I2SB | **38.1304** | 0.2168 | 23.0778 | 0.5957 | 37.1202 | 0.1782 | 24.2578 | 0.6492 |
| SDB (SB) | 38.1484 | **0.1936** | **23.7615** | 0.6293 | **36.6346** | **0.1569** | **24.9707** | 0.6811 |

Table 6: Motion deblurring performance on Flowers102 dataset using BTTB system matrix with Tikhonov-regularized pseudoinverse. Bold indicates best values.

| Method | FID↓ | LPIPS↓ | PSNR↑ | SSIM↑ |
|--------|------|--------|-------|-------|
| GOUB | 17.03 | 0.053 | 23.84 | 0.813 |
| IRSDE | 16.27 | 0.033 | 26.76 | 0.842 |
| DDBM | 16.00 | 0.040 | 29.74 | 0.892 |
| I2SB | 15.62 | **0.025** | 29.29 | 0.874 |
| SDB (SB) | **14.89** | **0.025** | **30.39** | **0.903** |

In practice, the initial estimate $\hat{\mathbf{x}}$ of the true signal $\mathbf{x}$ mentioned in appendix C.2 can be obtained via a few iterations of gradient descent on

$$\min_{\hat{\mathbf{x}}} \|\mathbf{A}(\hat{\mathbf{x}}) - \mathbf{y}\|_2^2,$$

providing a reasonable starting point for linearization. Steps 2 and 3 (appendix C.2) involve computing the Jacobian at the given point and using it to obtain the pseudoinverse reconstruction (PR) and range-nullspace projections for SDB. These operations can be implemented efficiently without explicitly storing the Jacobian, using Jacobian-vector products (JVPs) or gradient-based operations available in autodifferentiation frameworks such as PyTorch, or by exploiting an analytic expression for the Jacobian when available.

To illustrate the applicability of SDB to nonlinear systems, we perform a proof-of-concept benchmark on CIFAR10 [Krizhevsky et al., 2009] using a nonlinear contrast operator

$$\mathbf{A}(\mathbf{x}) = \sigma(k(\mathbf{x} - \boldsymbol{\alpha})),$$

where $\sigma$ is the sigmoid function, $k$ controls contrast strength, and $\boldsymbol{\alpha}$ is a contrast bias. We apply the linearization procedure described in appendix C.2: step 1 uses 5 iterations of gradient descent to obtain an initial guess, while steps 2 and 3 use an analytic Jacobian to compute the pseudoinverse reconstruction and projections.

As shown in table 7, SDB (SB) outperforms all supervised baselines across all metrics in this nonlinear contrast recovery task. This demonstrates that, even under nonlinear system responses, SDB can effectively leverage the locally linearized operator to produce high-fidelity reconstructions, achieving both superior perceptual quality (FID, LPIPS) and reconstruction accuracy (PSNR, SSIM). We emphasize that this represents only an initial proof-of-concept study, and future work should investigate the applicability of SDB to nonlinear systems more thoroughly.

### E.5 Hyperparameter analysis

We conduct an ablation study to analyze the sensitivity of SDB to its hyperparameters. Following prior works [Zhou et al., 2024], our stochastic process is undefined at $t = 1$, requiring sampling to

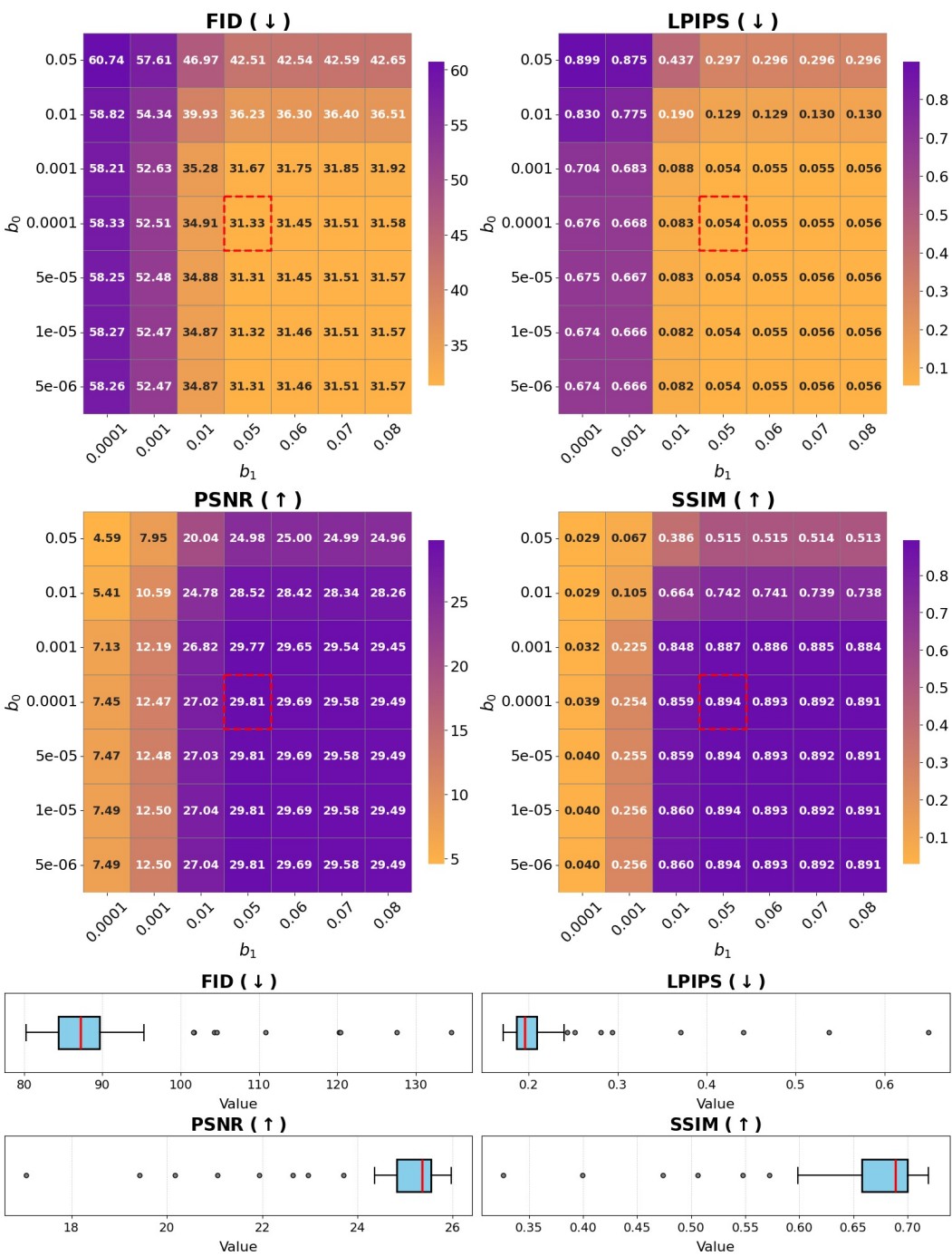

Figure 5: **Top**: Performance heatmaps for MRI reconstruction under varying $(\epsilon_1, \epsilon_2)$, using the optimal $(b_0, b_1)$ configuration. The red box highlights the LPIPS-optimal setting. **Bottom**: Boxplots showing the best achievable performance across 64 $(b_0, b_1)$ pairs on the superresolution task, where $(\epsilon_1, \epsilon_2)$ are tuned separately for each $(b_0, b_1)$ pair.

Table 7: Nonlinear contrast reconstruction on CIFAR10. Bold indicates best values.

| Method | FID↓ | LPIPS↓ | PSNR↑ | SSIM↑ |
|--------|------|--------|-------|-------|
| I2SB | 2.97 | 0.00103 | 37.53 | 0.9906 |
| GOUB | 4.60 | 0.00182 | 32.54 | 0.9743 |
| DDBM | 1.92 | 0.00016 | 42.74 | 0.9963 |
| IRSDE | 4.35 | 0.00140 | 33.53 | 0.9789 |
| SDB (SB) | **1.31** | **0.00015** | **44.48** | **0.9977** |

begin at $t = 1 - \epsilon_1$ for some $\epsilon_1 > 0$. Similarly, we terminate sampling at $t = \epsilon_2$, where $\epsilon_2 > 0$. For SDB (VP) and SDB (VE), $\epsilon_1$ and $\epsilon_2$ are the only hyperparameters. In contrast, SDB (SB) includes an additional design choice: the shape of the null-space diffusion coefficient $g(t)$. We parameterize it as

$$g^2(t) = \mathbb{1}_{t \leq 0.5} \hat{g}(t) + \mathbb{1}_{t > 0.5} \hat{g}(1 - t)$$

where $\hat{g}(t) = (\sqrt{b_0} + t(\sqrt{b_1} - \sqrt{b_0}))^2$, which is a continuous version of the coefficient proposed by Liu et al. [2023]. This introduces two additional hyperparameters, $b_0, b_1 > 0$. In the remainder of this section, we focus on SDB (SB), noting that we observed qualitatively similar trends for the VP and VE variants.

We first examine how the choice of $\epsilon_1$ and $\epsilon_2$ influences the optimal configuration of $(b_0, b_1)$ for SDB (SB) in the MRI reconstruction task. Figure 5 (top) shows the performance trend when varying $(\epsilon_1, \epsilon_2)$. We observe a simple and stable relationship with performance, where $\epsilon_1$ has a stronger influence than $\epsilon_2$. Moreover, a broad plateau emerges, allowing for easy tuning. We also highlight the best configuration in terms of LPIPS, which lies close to the optimal Pareto frontier in this setting.

Next, we analyze the stability of SDB (SB) across different $(b_0, b_1)$ configurations. Since these values determine the shape of the variance function, the optimal $\epsilon_1, \epsilon_2$ may differ across $(b_0, b_1)$ pairs. Figure 5 (bottom) presents boxplots of the best performance for 64 configurations of $(b_0, b_1)$ on the superresolution task. For each pair, we use grid search to identify the best $\epsilon_1, \epsilon_2$ values, leveraging the small size of the DIV2K dataset. The results reveal strong stability, with performance concentrated around the best observed value and only a few outliers.

### E.6 Ablation study on noise schedules

We evaluate the effect of different schedules for the VP and VE variants of SDB, considering linear, quadratic, and cosine versions on the superresolution task. Using our notation:

$$\alpha_t = 1 - t^2 \quad \text{(quadratic VP)}, \quad \alpha_t = \cos\left(\frac{\pi}{2}t\right) \quad \text{(cosine VP)},$$

$$\beta_t = \sigma_{\max} t^2 \quad \text{(quadratic VE)}, \quad \beta_t = \sigma_{\max} \sin\left(\frac{\pi}{2}t\right) \quad \text{(cosine VE)}.$$

We also assume $\gamma_t = \sigma_{\max}^{-1} \beta_t$ for VE.

Table 8: Ablation results for SDB (VP and VE) with different noise schedules. Lower FID and LPIPS indicate better perceptual quality; higher PSNR and SSIM indicate better reconstruction fidelity. Bold indicates the best value for each metric within VP or VE.

| Schedule | SDB (VP) | | | | SDB (VE) | | | |
|----------|---------|----------|---------|---------|---------|----------|---------|---------|
| | FID ↓ | LPIPS ↓ | PSNR ↑ | SSIM ↑ | FID ↓ | LPIPS ↓ | PSNR ↑ | SSIM ↑ |
| Linear | 87.08 | 0.228 | **25.91** | **0.724** | 94.73 | 0.226 | 25.90 | 0.718 |
| Quadratic | **75.25** | 0.174 | 24.77 | 0.680 | 76.88 | **0.159** | 25.29 | 0.678 |
| Cosine | 77.99 | **0.160** | 24.75 | 0.674 | **76.14** | 0.217 | **26.17** | **0.732** |

The results show a clear trade-off between perceptual quality (FID, LPIPS) and reconstruction fidelity (PSNR, SSIM). Linear schedules favor distortion metrics, while quadratic and cosine schedules improve perceptual metrics at the expense of reconstruction accuracy. In particular, Cosine SDB (VE) achieves the best values on three out of four metrics, demonstrating that exploring alternative schedules can further improve performance.

### E.7 Computational complexity and overhead of SDB

We analyze the computational complexity and runtime overhead of SDB compared to other supervised baselines. First, we discuss the asymptotic complexity, followed by empirical measurements.

**Complexity**

For the general case where the pseudoinverse can be computed analytically using a stored matrix $\mathbf{A}$, two cases arise: $m \geq d$ and $m < d$. The pseudoinverse can be computed as

$$\mathbf{A}^+ = (\mathbf{A}^\top \mathbf{A})^{-1} \mathbf{A}^\top \quad \text{or} \quad \mathbf{A}^+ = \mathbf{A}^\top (\mathbf{A} \mathbf{A}^\top)^{-1},$$

respectively. The resulting asymptotic complexities are summarized in table 9. Computing the PR has the same complexity (in the $O$ sense) as computing the pseudoinverse itself.

Table 9: Asymptotic computational complexity for pseudoinverse computation and range/null space projections.

| Operation Type | $m \geq d$ | $m < d$ |
|---|---|---|
| Pseudoinverse | $O(md^2 + d^3)$ | $O(m^2 d + m^3)$ |
| Range Projection $\mathbf{A}\mathbf{A}^+$ | $O(md^2)$ | $O(m^2 d)$ |
| Null Projection $(\mathbf{I} - \mathbf{A}^+ \mathbf{A})$ | $O(d^2 m)$ | $O(dm^2)$ |

In practice, we do not explicitly construct $\mathbf{A}^+$. Instead, we implement the application of $\mathbf{A}$, $\mathbf{A}^\top$, $\mathbf{A}^+$, and their combinations to obtain projections and PRs. For each inverse problem, these implementations differ. For example, in inpainting, $\mathbf{A}$ corresponds to element-wise masking, whereas in CT reconstruction, truncated SVD is used to implement $\mathbf{A}$ efficiently, typically resulting in lower empirical complexity.

**Empirical Overhead**

We measured runtime in seconds for the most computationally intensive operations on the CT reconstruction task. The evaluation was conducted over four batches of size eight. We report three types of operations:

- Forward Step Without Network: sampling from $p(\mathbf{x}_t \mid \mathbf{x}_0)$,
- Forward Step With Network: Forward Step Without Network plus a forward pass through the neural network,
- Reverse Step With Network: sampling from $p(\mathbf{x}_{t-\Delta t} \mid \mathbf{x}_t, \mathbf{x}_1)$ using the neural network, corresponding to a single discrete reverse-time step.

The empirical runtimes are summarized in table 10.

Table 10: Runtime of SDB and baseline supervised methods for the CT reconstruction task (seconds per batch of size 8).

| Method | Forward Step Without Network | Forward Step With Network | Reverse Step With Network |
|---|---|---|---|
| SDB (SB) | 0.0062 | 0.0885 | 0.5981 |
| I2SB | 0.0003 | 0.0821 | 0.5851 |
| IR-SDE | 0.0003 | 0.0832 | 0.5895 |
| GOUB | 0.0008 | 0.0830 | 0.5863 |
| DDBM | 0.0011 | 0.0825 | 0.5880 |

While Forward Step With Network shows a small increase in computation time for SDB compared with other methods, this overhead is negligible in practice. The relative cost of the neural network forward pass dominates runtime, and matrix-based operations are efficiently parallelized on the NVIDIA A100 GPU used for all experiments. Consequently, SDB runtime during inference is nearly identical to that of other supervised methods.

### E.8 OT unpaired-data-based baselines

We provide an additional small-scale comparison with recent OT-based unpaired-data methods (UNSB by Kim et al. [2024] and SBF by De Bortoli et al. [2024]) on the MRI reconstruction task, as shown in table 11. Following the original training procedures, we trained the models to map between the distributions of PRs and true signal observations. Since these methods lack direct supervision from paired data, their performance is naturally inferior to that of supervised approaches.

Table 11: MRI reconstruction results across all supervised and unpaired-data-based methods. Lower FID and LPIPS indicate better perceptual quality; higher PSNR and SSIM indicate better reconstruction fidelity.

| Metric | Unpaired-data-based | | Supervised | | | | | | |
|---|---|---|---|---|---|---|---|---|---|
| | UNSB | SBF | I2SB | IR-SDE | GOUB | DDBM | SDB (VP) | SDB (VE) | SDB (SB) |
| FID ↓ | 35.91 | 43.63 | 31.54 | 30.14 | 30.63 | 32.42 | 32.88 | 33.90 | **29.85** |
| LPIPS ↓ | 0.237 | 0.290 | 0.065 | 0.065 | 0.058 | 0.074 | 0.068 | 0.083 | **0.053** |
| PSNR ↑ | 18.72 | 18.46 | 28.75 | 28.88 | 28.59 | 28.97 | 29.26 | 29.10 | **29.81** |
| SSIM ↑ | 0.483 | 0.291 | 0.849 | 0.871 | 0.863 | 0.872 | 0.881 | 0.876 | **0.893** |

# F Extended discussion

## F.1 Advantages of Embedding Measurement System Information

Incorporating measurement system information into the generative process offers several conceptual and practical advantages. While earlier approaches that relied on hand-crafted priors often provided limited gains, the proposed SDB framework does not depend on such priors. Instead, SDB embeds the measurement system directly into the model dynamics, which naturally decomposes the inverse problem into two orthogonal components with distinct roles: the range space and the null space. This decomposition provides a structured way to separate reconstruction from synthesis without manually injecting prior knowledge.

By embedding the measurement operator, SDB defines a more efficient stochastic path from the observed measurements to the clean image, akin to other diffusion bridge models that refine conditional diffusion by employing more informed reverse-process initializations. Within this framework, the neural network simultaneously learns two complementary tasks: (i) an optional denoising task in the range space—typically simpler and often directly constrained by the data—and (ii) a synthesis task in the null space, which accounts for missing or unobserved components.

In noise-free scenarios, this structure allows exact preservation of range-space content, ensuring that information already supported by the measurements remains unaltered. Under noisy conditions, the model benefits from the fact that the range space already encodes the correct underlying structure, requiring only localized refinement. Consequently, the range space acts as a structurally informative prior that guides the null-space synthesis. For example, in generative inpainting tasks—where the unmasked region is noise-free or nearly so—SDB avoids unnecessary corruption of the observed region and concentrates its modeling capacity on generating the missing content.

## F.2 Perception-Distortion Tradeoff

The trade-off between perception and distortion mentioned by Blau and Michaeli [2018] is an important topic in the context of this work. Thanks to a specific formalization, where perception is measured as a divergence between the true distribution and the one induced by the model, while distortion is quantified through the expected reconstruction error, one may hypothesize about the behavior of a given algorithm from a theoretical point of view. Following the formulation in Blau and Michaeli [2018], since perception and distortion are related through a Pareto front, one can draw more principled conclusions from quantitative results.

The range-nullspace decomposition provided by the assumed system response matrix also enables deeper insights. In comparison to other methods, SDB's advantage lies in its differential treatment of the range and null spaces: it must only (optionally) denoise the former, while new content is

synthesized in the latter. This dual treatment leads to improved performance from the outset. In the noise-free case, the range space component is left untouched, resulting in zero distortion, while in the noisy case it undergoes milder transformations to reach the final result, which intuitively should also reduce distortion. Under a noise-free scenario, it is also easier to reason about perception: if the model synthesizes null-space content that perfectly aligns with the ground truth image, it achieves zero distortion at an inevitable cost in perception. Conversely, if it synthesizes null-space content that is fully in-distribution but deviates from the true signal, perception is maximized at the cost of distortion.

From a theoretical perspective, in the limit, SDB samples from the true conditional distribution $p(\mathbf{x} \mid \mathbf{y})$, which suggests that the method is inclined toward perfect perception. However, since we implicitly assume a non-empty null space, SDB will always lack some information required to perfectly reconstruct the true $\mathbf{x}$, mirroring the trade-off considered by Blau and Michaeli [2018].

In practice, we observe a clear trend of SDB outperforming all other methods in three out of four metrics in the super-resolution task (table 2), while obtaining slightly worse results in terms of LPIPS. Following Blau and Michaeli [2018], this observation leads to an interesting insight. Better FID, representing a divergence between distributions, indicates that SDB achieves better perceptual results. However, it is also superior in terms of PSNR and SSIM, which measure reconstruction accuracy (distortion). Based on Blau and Michaeli [2018], one could argue that, in this task, the methods operate near the optimal performance. Since LPIPS does not quantify distributional divergence but instead measures perceptual distortion in the space of neural network representations, it illustrates how a specific type of semantic reconstruction must be sacrificed to move closer to the Pareto front. Moreover, the ablation study on noise schedulers for the VP and VE variants of SDB shows how better LPIPS values can be recovered, providing another manifestation of this trade-off.

## G   Broader impact

By incorporating measurement system parameters into its SDE, SDB offers positive societal impacts by enabling more accurate and efficient reconstructions in critical applications such as medical imaging, remote sensing, and scientific inverse problems, thereby supporting improved diagnostics, sustainability, and broader accessibility. However, it also poses potential negative societal risks, including misuse for surveillance or de-anonymization, amplification of bias from flawed measurement models, and overreliance on plausible but incorrect outputs in high-stakes domains. Additionally, the method could be repurposed for generating deceptive content from limited data. However, SDB and prior bridge methods have not been demonstrated at industry-scale deployment levels, and in practice, they are typically suited for problem-specific applications rather than large-scale settings with vast amounts of data. This limits their potential for misuse in broader societal contexts.

## H   Additional visual results

We provide additional qualitative samples for inpainting (fig. 6), superresolution (fig. 7), MRI reconstruction (fig. 8) and CT reconstruction (fig. 9).

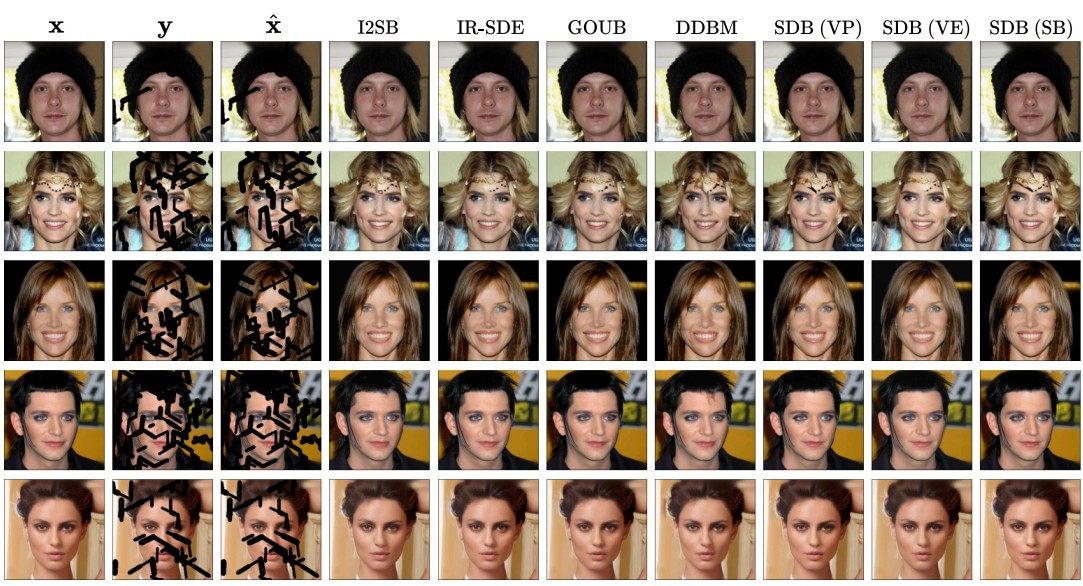

Figure 6: Qualitative comparison of SDB variants with the best-performing baselines (bridge methods). Rows depict the results for inpainting.

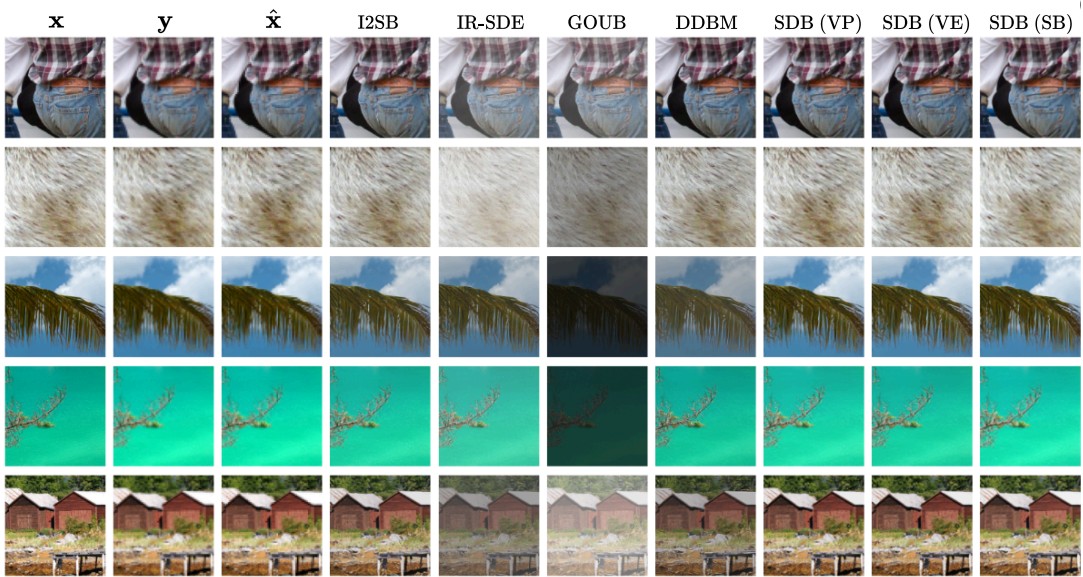

Figure 7: Qualitative comparison of SDB variants with the best-performing baselines (bridge methods). Rows depict the results for superresolution.

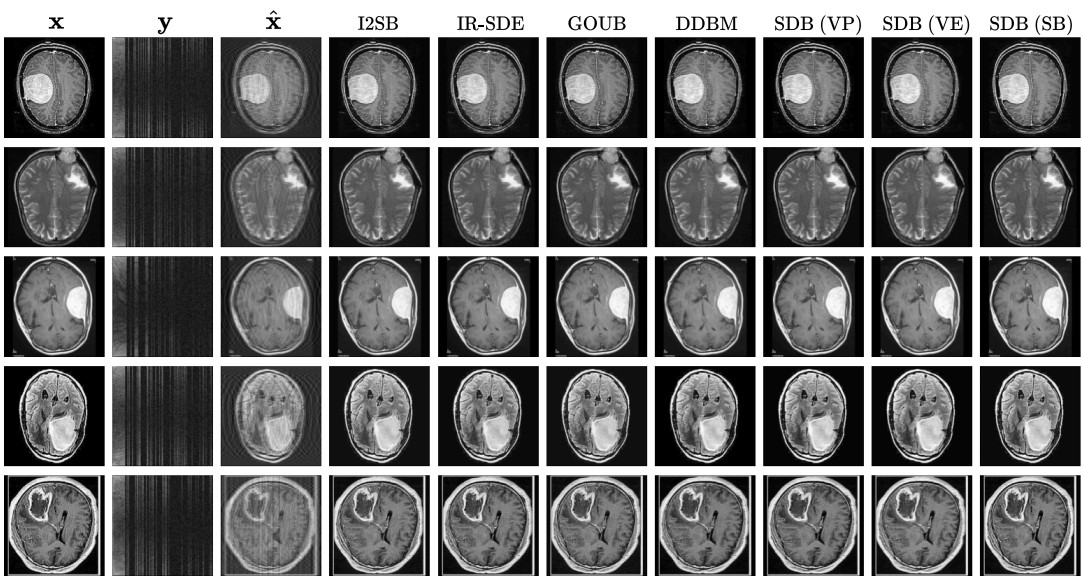

Figure 8: Qualitative comparison of SDB variants with the best-performing baselines (bridge methods). Rows depict the results for MRI reconstruction.

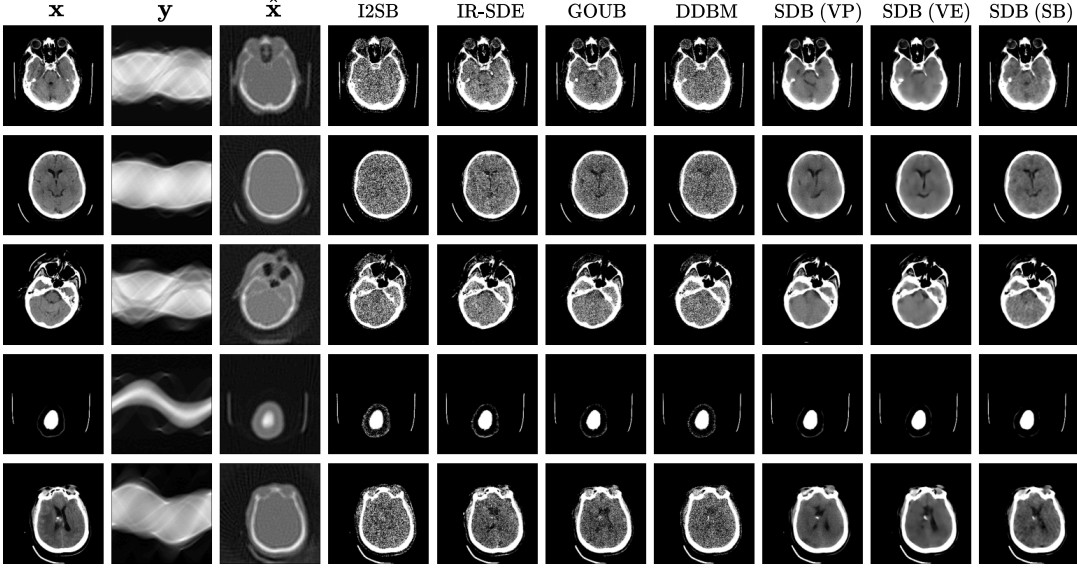

Figure 9: Qualitative comparison of SDB variants with the best-performing baselines (bridge methods). Rows depict the results for CT reconstruction.

