# OpenReview forum: "System-Embedded Diffusion Bridge Models"
_NeurIPS.cc/2025/Conference — NeurIPS 2025 poster_

### Official Review · Reviewer_JgTN · 2025-06-19

**Clarity:** 4
**Significance:** 3
**Originality:** 3
**Rating:** 5
**Confidence:** 4

**Summary:**

This work identifies that many existing supervised methods for inverse problem solving ignore the a priori knowledge of the degradation operator. Motivated by this findings, the authors develop a new type of Schrödinger bridge model that takes the degradation operator into account for choosing the form of the SDE applied in training. In several experiments, the authors demonstrate the superiority of their approach over previous methods.

**Questions:**

Please see weaknesses.
Additionally:
* Could the proposed method be applied such that several degradations are handled by a single model?

**Ethical Concerns:**

["NO or VERY MINOR ethics concerns only"]

**Final Justification:**

I believe this an insightful paper with a novel approach and should be accepted. While the concerns I have raised have only been partially addressed by the rebuttal, I believe that the paper is of more than sufficient quality for acceptance.

**Limitations:**

Yes

**Quality:**

4

**Strengths And Weaknesses:**

Strengths:
* The paper’s motivation is clearly established, and the writing is easy to follow. I find that system-embedded diffusion bridges are an insightful extension to Schrödinger bridge models, making the most of the a priori knowledge of the inverse problem.
* The evaluations are sound, and the results shown in the paper are encouraging. Moreover, SDB’s generalization capabilities are non-trivial.

Weaknesses:
* While the empirical results clearly show an improvement, I believe the manuscript lacks a theoretical analysis, hypothesis or discussion for why methods that embed the measurement information into the sampling scheme are desired. Many fields in deep learning have shown that hand-crafted priors may offer only limited benefits over systems that purely model the training data, for fully converged networks with sufficient data. The authors should take extra care to demonstrate that the increase in performance is solely driven by the measurement-embedding technique, and disentangle all other differences.
* I believe that a discussion of the Perception-Distortion Tradeoff [1] is warranted for the comparisons made in this work.
* Minor: I believe the paper would be easier to follow if a training and sampling algorithm was included.

[1] Blau, Yochai, and Tomer Michaeli. "The perception-distortion tradeoff." Proceedings of the IEEE conference on computer vision and pattern recognition. 2018.

---

> ### Author Rebuttal · Authors · 2025-07-31
>
> We are grateful to the reviewer for the insightful feedback and such appreciation of our work. We aim to address the concerns with the comments below.
>
> ## (W1.) Discussion on embedding the system information
>
> Thank you for pointing out the lack of discussion on why methods that incorporate measurement system information should be preferred. While some earlier approaches based on hand-crafted priors offered limited benefits, SDB does not rely on such priors. Instead, it embeds the measurement system directly, which disentangles the problem into two orthogonal components with distinct characteristics—range and null spaces—without injecting hand-crafted information. This embedding defines a more efficient stochastic path from measurements to clean images, similar in motivation to other diffusion bridge models, which improve upon standard conditional diffusion by using more informed reverse-process initializations. In SDB, the neural network gains principled access to an (optional) denoising task in the range space—typically simpler—and a more difficult synthesis task in the null space. In noise-free settings, this enables exact preservation of the range space content; in noisy settings, the model is guided by the fact that the range space already contains the correct structure, requiring only refinement. In both cases, the range space provides a structurally informative prior for null space synthesis. For example, in generative inpainting, where the unmasked region has low or zero noise, SDB avoids unnecessary corruption of that region and allows the model to focus its capacity on generating only the missing content.
>
> We agree that making claims about the influence of the novel matrix-valued parameterization must be supported with a careful disentanglement of other factors. In the paper, we attempted to do that by
> i) retraining all methods (both supervised and unsupervised) from scratch ii) using the same network architecture and iii) training procedure, including optimizers, learning rate schedulers and their respective hyperparameters. Moreover, we provide a general implementation that unifies all of the methods within a single framework, supporting reproducibility and fairer comparisons, e.g., in terms of runtime. We also conducted a broad hyperparameter analysis for the SB variant in Appendix E.2. and include additional ablations regarding VP and VE variants to further disentangle their particular influence on performance.
>
> ### Ablation study on linear, quadratic and cosine noise schedules
>
> Using the parameterization introduced in our paper, we select $\alpha_t = 1 - t^2$ for the quadratic scheduler and $\alpha_t = \cos\left(\frac{\pi}{2} t\right)$ for the cosine scheduler for the VP variant. In the variance-exploding (VE) case, we let $\beta_t = \sigma_{\text{max}} t^2$ for the quadratic schedule, and $\beta_t = \sigma_{\text{max}} \sin\left(\frac{\pi}{2} t\right)$ for the cosine version. The remaining terms are derived as in lines 202–203 of our submission, assuming the standard relationship $\gamma_t = \sigma_{\text{max}}^{-1} \beta_t$ in the VE setting.
>
> #### SDB (VP)
>
> |Scheduler|FID↓|LPIPS↓|PSNR↑|SSIM↑|
> |-|-|-|-|-|
> |Linear|87.08|0.228|**25.91**|**0.724**|
> |Quadratic|**75.25**|0.174|24.77|0.680|
> |Cosine|77.99|**0.160**|24.75|0.674|
>
> #### SDB (VE)
>
> |Scheduler|FID↓|LPIPS↓|PSNR↑|SSIM↑|
> |-|-|-|-|-|
> |Linear|94.73|0.226|25.90|0.718|
> |Quadratic|76.88|**0.159**|25.29|0.678|
> |Cosine|**76.14**|0.217|**26.17**|**0.732**|
>
> These findings reveal that the use of more advanced scheduling strategies alters the balance between perceptual quality (FID, LPIPS) and reconstruction accuracy (PSNR, SSIM). The linear scheduler tends to favor distortion-based metrics, whereas both quadratic and cosine variants emphasize perceptual improvements, even at the expense of some distortion. Notably, the cosine SDB (VE) configuration surpasses all previous baselines and SDB setups in three out of four metrics. This underlines the potential value of deeper investigations into the dynamics of SDB’s underlying stochastic process.
>
> ## (W2.) Discussion of the Perception-Distortion Tradeoff
>
> We agree that the trade-off between perception and distortion mentioned by [1] is an important avenue for discussion in our context. Thanks to a specific formalization, where perception is measured as a divergence between the true distribution and the one induced by the model, while distortion is quantified through expected reconstruction error, one may hypothesize about the behavior of a given algorithm from a theoretical point of view. Following the paper, since perception and distortion are related through a Pareto front, one can draw more principled conclusions from the quantitative results.
>
> In fact, the range-nullspace decomposition provided by the assumed system response matrix also allows for obtaining deeper insights. In comparison to other methods, SDB's advantage is that it treats the range and null spaces differently -- it must only (optionally) denoise the former, while new content is only synthesized in the latter. This dual treatment should provide improved performance from the outset -- in the noise-free case, the range space component is left untouched, leading to its zeroed distortion, while in the noisy case it undergoes milder transformations to arrive at the final result, which intuitively should also lower the distortion. Under a noise-free scenario, it is also easier to reason about perception -- if the model synthesizes null-space content that is perfectly aligned with the ground truth image, it will achieve zero distortion at an inevitable cost in perception. On the other hand, if it always synthesizes null-space content that is fully in-distribution, but deviates from the true signal, perception is maximized at the cost of distortion.
>
> One may also consider more theoretical aspects -- as shown in the response to Reviewer unSw (**(W3.)**), in the limit, SDB samples from the true conditional distribution $p(\mathbf{x}\mid\mathbf{y})$, which suggests that the method is inclined toward perfect perception. On the other hand, since we implicitly assume a non-empty null space, SDB will always lack some information to perfectly reconstruct the true $\mathbf{x}$, mirroring the trade-off considered in [1].
>
> In practice, we observe an evident trend of SDB outperforming all other methods in 3 out of 4 metrics in the superresolution task (Table 2 in the main paper), while obtaining slightly worse results in terms of LPIPS. Following [1], this observation leads to an interesting insight. Better FID, representing a form divergence between distributions, means that SDB obtains better perceptual result. However, it is also superior in terms of PSNR and SSIM, which measure reconstruction accuracy (distortion). Based on [1], one could argue that, in this task, the methods are close to the optimal performance -- since LPIPS does not quantify distributional divergence, but instead measures perceptual distortion in the space of neural network representations, it shows how a specific type of "reconstruction" (in this case: semantic, human-understandable) must be sacrificed to obtain results closer to the Pareto front. Moreover, the above ablation study on the noise schedulers for VP and VE variants of SDB shows how better values of LPIPS can be recovered, providing another form of trade-off.
>
> We would be happy to include such extended discussion in the updated version of our work.
> ## (W3.) Training and sampling algorithms
> Due to space constraints, these algorithms were included in Appendix C.3 of the original submission. We would be happy to move them into the main text in a revised version of the paper.
>
> ## (Q1.) Several degradations handled by a single model
> Thank you for raising this point. Indeed, SDB can handle multiple degradations within a single model. This is supported by the fact that a cascade of multiple linear systems with additive Gaussian noise can always be modelled by an equivalent total linear system and total additive Gaussian noise. For example, in the MRI reconstruction task, the measurement system combines a Fourier transform followed by a spatial-frequency-domain masking operator. As Table 2 shows, SDB outperforms all baselines for that task. A formal theorem and proof to support this claim is provided below.
>
> ### Theorem
> Let the random variables be defined recursively:
> - $ \mathbf{y}_1 \sim \mathcal{N}(\mathbf{A}_1 \mathbf{x}, \boldsymbol{\Sigma}_1) $
> - $ \mathbf{y}_2 \sim \mathcal{N}(\mathbf{A}_2 \mathbf{y}_1, \boldsymbol{\Sigma}_2) $
> - $\ldots$
> - $ \mathbf{y}_k \sim \mathcal{N}(\mathbf{A}_k \mathbf{y}ₖ₋₁, \boldsymbol{\Sigma}_k) $
>
> Then the marginal distribution of $ \mathbf{y}_k $ given $ \mathbf{x} $ is:
>
> $\mathbf{y}_k \sim \mathcal{N} \big(\mathbf{A}_k \cdots \mathbf{A}_1 \mathbf{x}, \boldsymbol{\Omega}_k \big),$
>
> where the covariance satisfies
>
> $\boldsymbol{\Omega}_1 = \boldsymbol{\Sigma}_1,$
>
> $\boldsymbol{\Omega}_j = \mathbf{A}_j \boldsymbol{\Omega}ⱼ₋₁ \mathbf{A}_j^\top + \boldsymbol{\Sigma}_j,$
>
> for $j=2,\ldots,k$.
>
> ### Proof
> By induction on $k$. For $k=1$, the claim holds by definition. Assume it holds for $j$, then
>
> $\mathbf{y}ⱼ₊₁ = \mathbf{A}ⱼ₊₁ \mathbf{y}_j + \boldsymbol{\epsilon}ⱼ₊₁,$
>
> $\boldsymbol{\epsilon}ⱼ₊₁ \sim \mathcal{N}(0, \boldsymbol{\Sigma}ⱼ₊₁).$
>
> Since $\mathbf{y}_j$ is Gaussian and independent of $\boldsymbol{\epsilon}ⱼ₊₁$,
>
> $$
> \mathbf{y}ⱼ₊₁ \sim \mathcal{N}\left(\mathbf{A}ⱼ₊₁ \mathbb{E}[\mathbf{y}_j], \mathbf{A}ⱼ₊₁ \boldsymbol{\Omega}_j \mathbf{A}ⱼ₊₁^\top + \boldsymbol{\Sigma}ⱼ₊₁\right),
> $$
>
> and the mean and covariance match the induction step, completing the proof.
>
> ## Summary
>
> We hope these clarifications address the concerns and are open to further discussion regarding any other issues.
>
> ---
>
> [1] Blau, Yochai, and Tomer Michaeli. "The perception-distortion tradeoff." Proceedings of the IEEE conference on computer vision and pattern recognition. 2018.

---

> > ### Comment · Reviewer_JgTN · 2025-08-05
> >
> > I thank the authors for their rebuttal. My concerns have either been partially addressed, or promised to be addressed in the final version.
> > I maintain my recommendation to accept.

---

> ### Author Response · Authors · 2025-08-05
>
> Dear Reviewer JgTN,
>
> We sincerely thank you for taking the time to evaluate our submission. We would be glad to discuss any aspect of the work further and respond to any concerns that may arise.

---

> ### Author Response · Authors · 2025-08-08
>
> Dear Reviewer JgTN,
>
> Thank you for your involvement in the review process and the discussion phase, as well as for recognizing the novelty of our work; we truly appreciate your engagement. We will incorporate your recommendations into the revised version of the paper.

---

### Official Review · Reviewer_FGzg · 2025-06-19

**Clarity:** 3
**Significance:** 3
**Originality:** 3
**Rating:** 4
**Confidence:** 4

**Summary:**

This paper introduces System-embedded Diffusion Bridge Models (SDBs), a framework for supervised diffusion bridge methods that explicitly embeds the known linear measurement system into the coefficients of matrix-valued stochastic differential equations. The key contribution lies in leveraging the range-nullspace decomposition of linear operators to design specialized diffusion processes that handle range and null space components differently. The authors propose three variants (SDB-SB, SDB-VP, SDB-VE) and demonstrate improvements over existing bridge methods (I2SB, IR-SDE, GOUB, DDBM) across four inverse problems: image inpainting, super-resolution, CT reconstruction, and MRI reconstruction.

**Questions:**

I would like to know following the cons:

1. Thm1 applicability. The Magnus expansion approximation becomes exact when drift matrices commute. How restrictive is this condition for the proposed SDB variants? Do the specific choices of parameters in equations (8a-8b) satisfy this condition?
2. Pseudoinverse stability. For rank-deficient measurement matrices A, how does numerical stability of the pseudoinverse computation affect SDB performance? Have the authors considered regularized pseudoinverses for ill-conditioned systems?
3. What is the computational complexity of the matrix-valued SDE compared to scalar formulations? How does this scale with problem dimension d and measurement dimension m?
4. The authors mention local linearization as a potential workaround for nonlinear measurement systems. Can you provide more details on how this would work in practice and its limitations? e.g., failure cases
5. The current formulation treats range and null spaces independently. Could there be benefits to introducing coupling terms between these spaces?

**Ethical Concerns:**

["NO or VERY MINOR ethics concerns only"]

**Final Justification:**

I thank the authors for their detailed response and effort on the results. they had addressed my concerns. hope authors can revise it in the final version accordingly. I maintain my positive rating.

**Limitations:**

please refer to the questions and cons.

**Paper Formatting Concerns:**

please double check the inconsistent matrix notation (e.g., A) and vector notations (e.g., x), and references. Table 1 is bit difficult to read

**Quality:**

3

**Strengths And Weaknesses:**

pros:
1. The principled embedding of measurement systems into matrix-valued SDE coefficients represents a theoretical advancement beyond scalar SDEs. The connection to optimal transport theory through Theorem 2 provides solid mathematical foundation.
2. The paper evaluates across diverse inverse problems with varying system complexities, from simple inpainting to complex medical imaging tasks (CT/MRI reconstruction), demonstrating broad applicability.
3. The systematic evaluation under system misspecification (Section 5.2) addresses a critical practical concern, showing SDB's superior generalization when training and deployment parameters differ.
4.  The range-nullspace decomposition provides intuitive understanding of how SDB handles known (range) and missing (null space) information differently, which is conceptually appealing for inverse problems.
4. SDB consistently outperforms baselines across metrics, with particularly notable improvements in medical imaging tasks where the gains are substantial.

cons:
1. While Theorem 1 from Tivnan et al. [2025] provides the foundation, the paper lacks theory analysis of the novel matrix-valued formulation. The conditions under which the Magnus expansion approximation becomes equality (commuting matrices) are restrictive and not proper discussed.
2. The framework is limited to linear Gaussian measurement models. While the authors acknowledge this limitation, many real-world inverse problems involve nonlinear degradations that cannot be addressed by this approach.
3. The paper does not analyze the computational overhead of the matrix-valued SDE formulation compared to scalar methods. The range-nullspace decomposition and pseudoinverse computations may introduce significant computational costs and bias.
4. While the paper compares against recent bridge methods, it lacks comparison with state-of-the-art unsupervised methods like DPS, πGDM variants, or recent optimal transport approaches that don't require paired training data.
5. The authors use simple linear scheduling for range and null space coefficients, missing opportunities for more sophisticated designs that could better leverage the dual-process structure.

---

> ### Author Rebuttal · Authors · 2025-07-31
>
> We thank the reviewer for their insightful feedback. Below we address the noted weaknesses and questions.
>
> ## (W1., Q1.) Applicability of Theorem 1, restrictiveness of the Magnus expansion
>
> As noted, the matrix integral form holds if the drift matrices $F_t$ commute for all $t\in[0,1]$. This condition is always satisfied for SDB under Eq. (10a): the only time-dependent term is scalar $\alpha_t$, so $F_t$ has constant singular vectors and thus commutes for all $t$. This applies to all scalar diffusion processes (e.g., VP, VE, Flow Matching OT [1], etc.), Fourier Diffusion [2], and Subspace Diffusion [4] models. We are not aware of any methods with non-commuting $F_t$, but even if present, Theorem 1 still applies via the Magnus expansion [3].
>
> ## (Q2.) Influence of pseudoinverse stability
>
> We address pseudoinverse stability through both the CT reconstruction task (with a misspecified setting from the main manuscript) and a new motion deblurring benchmark using Tikhonov regularization.
>
> ### Truncated SVD in CT reconstruction
>
> The system matrix models line integrals in CT, with SVD precomputed and used in all operations. Truncated SVD is standard for regularization—zeroing out low singular values linked to noise/high frequencies. The $\tau$ threshold controls its strength; during training we set $\tau=3.2$ to simulate low-dose scenarios. As shown in Table 2, SDB consistently outperforms all baselines, showing robustness to regularization. These findings are reinforced in the misspecified setting (Appendix E.1, Fig. 4), where increasing $\tau$ only strengthens SDB’s relative performance.
>
> ### Tikhonov regularization in motion deblurring
>
> To explore a distinct ill-posed case, we include a benchmark on 128×128 flower images (Flowers102 dataset [5]), following Reviewer unSw’s suggestion. The system matrix is a BTTB (block Toeplitz with Toeplitz blocks) from 2D convolutions. We apply the pseudoinverse via zero-order Tikhonov regularization (Wiener filtering) in the frequency domain whenever needed (e.g., range projection). Despite approximation error, SDB matches the best LPIPS and surpasses all supervised baselines in FID, PSNR, and SSIM:
>
> |Method|FID↓|LPIPS↓|PSNR↑|SSIM↑|
> |-|-|-|-|-|
> |GOUB|17.03|0.053|23.84|0.813|
> |IRSDE|16.27|0.033|26.76|0.842|
> |DDBM|16.00|0.040|29.74|0.892|
> |I2SB|15.62|**0.025**|29.29|0.874|
> |SDB (SB)|**14.89**|**0.025**|**30.39**|**0.903**|
>
> ### Summary
>
> These results demonstrate SDB’s resilience to pseudoinverse instability. We’re happy to include further details (e.g., on Flowers102 training) in the main text or Appendix.
>
> ## (W2., Q4.) Application of SDB to nonlinear systems via local linearization
>
> We outline how SDB applies to nonlinear systems via local linearization, then demonstrate this with a proof-of-concept experiment on nonlinear contrast in CIFAR10. Although SDB was not originally designed for nonlinear problems, the initial results demonstrate strong effectiveness.
>
> ### Theory
>
> SDB assumes a global range-nullspace decomposition of $\mathbf{A}$, valid in linear cases but not nonlinear. For a differentiable nonlinear $\mathbf{A}$, we approximate it via a first-order Taylor expansion:
>
> $$\mathbf{A}(\mathbf{x}) \approx \mathbf{A}(\mathbf{x}_0) + \mathbf{J}_A(\mathbf{x}_0)(\mathbf{x} - \mathbf{x}_0) \tag{1}$$
>
> with $\mathbf{x}_0$ near $\mathbf{x}$. The Jacobian $\mathbf{J}_A$ defines a local linearization of $\mathbf{A}$, enabling pseudoinverse and range/nullspace projections for SDB in a global sense, but at a natural cost of inaccuracies. Theoretically, the full procedure is:
> 1. Given $\mathbf{y} = \mathbf{A}(\mathbf{x}) + \boldsymbol{\Sigma}^{1/2}\epsilon$, estimate $\mathbf{x}$ with $\hat{\mathbf{x}}$ via maximum likelihood.
> 2. Linearize $\mathbf{A}$ at $\hat{\mathbf{x}}$, get $\mathbf{J}_A(\hat{\mathbf{x}})$.
> 3. Approximate $\mathbf{y} \approx \mathbf{J}_A(\hat{\mathbf{x}})\mathbf{x} + \boldsymbol{\Sigma}^{1/2}\epsilon$, which forms a linear Gaussian model suitable for SDB.
>
> ### Practice
>
> In practice, step 1 uses a few gradient descent steps on $\min_{\hat{\mathbf{x}}} || \mathbf{A}(\hat{\mathbf{x}}) - \mathbf{y} ||_2$ for an initial guess. Steps 2–3 use Jacobian-vector products (JVPs) or gradients—easily available via autodiff (e.g., PyTorch)—or analytic Jacobians if possible.
>
> To test this, we benchmark nonlinear contrast on CIFAR10 [8], with $\mathbf{A}(\mathbf{x}) = \sigma(k(\mathbf{x} - \alpha))$, where $\sigma$ is sigmoid, $k$ controls contrast, and $\alpha$ is bias. We apply our linearization steps: step 1 uses 5 gradient descent steps, and steps 2–3 use the analytic Jacobian.
>
> SDB outperforms all baselines under this nonlinear setup:
>
> |Method|FID↓|LPIPS↓|PSNR↑|SSIM↑|
> |-|-|-|-|-|
> |I2SB|2.97|0.00103|37.53|0.9906|
> |GOUB|4.60|0.00182|32.54|0.9743|
> |DDBM|1.92|0.00016|42.74|0.9963|
> |IRSDE|4.35|0.00140|33.53|0.9789|
> |SDB (SB)|**1.31**|**0.00015**|**44.48**|**0.9977**|
>
> ### Summary
>
> This provides early evidence that SDB can handle nonlinear problems, which we view as a promising direction for future work.
>
> ## (W3., Q3.) Computational overhead and complexity of SDB
>
> We first discuss the asymptotic complexity, then show that SDB’s overhead is negligible in practice.
>
> ### Complexity
>
> If $\mathbf{A}$ is stored in memory and $\mathbf{A}^+$ is computed analytically, we distinguish two cases: $m \geq d$ and $m < d$. Then, $\mathbf{A}^+ = (\mathbf{A}^\top \mathbf{A})^{-1} \mathbf{A}^\top$ or $\mathbf{A}^+ = \mathbf{A}^\top (\mathbf{A} \mathbf{A}^\top)^{-1}$ respectively. The resulting complexities:
>
> |Operation|$m \geq d$|$m < d$|
> |-|-|-|
> |Pseudoinverse|$O(md^2 + d^3)$|$O(m^2d + m^3)$|
> |Range Projection $\mathbf{A}\mathbf{A}^+$|$O(md^2)$|$O(m^2d)$|
> |Null Projection $(\mathbf{I} - \mathbf{A}^+\mathbf{A})$|$O(d^2m)$|$O(dm^2)$|
>
> In practice, we don’t form $\mathbf{A}^+$ explicitly, but use its actions via matrix-vector operations. These vary by task: e.g., inpainting uses a binary mask, CT uses truncated SVD. Actual runtime is often lower than theoretical bounds.
>
> ### Overhead
>
> We report runtime (in seconds) on CT reconstruction—the most demanding setting—averaged over 4 batches (size 8, A100 GPU). Three steps are shown: **Forward Step Without Network** (sampling $p(\mathbf{x}_t|\mathbf{x}_0)$), **Forward With Network** (adds NN forward pass), and **Reverse With Network** (a single reverse-time step).
>
> |Method|Forward w/o Net (s)|Forward w/ Net (s)|Reverse w/ Net (s)|
> |-|-|-|-|
> |SDB (SB)|0.0062|0.0885|0.5981|
> |I2SB|0.0003|0.0821|0.5851|
> |IRSDE|0.0003|0.0832|0.5895|
> |GOUB|0.0008|0.0830|0.5863|
> |DDBM|0.0011|0.0825|0.5880|
>
> While projections increase **Forward w/o Net** time, the **Forward w/ Net** and **Reverse** steps are dominated by network inference. Thanks to CUDA parallelism on A100 GPUs, SDB’s overall runtime closely matches that of other methods.
>
> ## (W4.) Comparison with SOTA baselines
>
> We note that both DPS and $\Pi\text{GDM}$ are already included in Table 2 of the main manuscript.
>
> ### Two new optimal transport baselines (unpaired)
>
> We thank the reviewer for suggesting unpaired OT methods. While typically absent from bridge literature, we include two: Unpaired Neural Schrödinger Bridge (UNSB) [6] and Schrödinger Bridge Flow (SBF) [7]. We trained both on MRI reconstruction (between PRs and true signals). Though unsupervised, they perform competitively:
>
> |Method|FID↓|LPIPS↓|PSNR↑|SSIM↑|
> |-|-|-|-|-|
> |UNSB|35.91|0.237|18.72|0.483|
> |SBF|43.63|0.290|18.46|0.291|
>
> Due to time limits, we couldn’t run all datasets but will include full results in the paper.
>
> ## (W5.) More sophisticated coefficient design
> The SB variant of SDB already uses nonlinear coefficient scheduling; Appendix E.2 includes extensive ablations on its shape and truncation.
>
> ### New ablation: linear, quadratic, cosine scheduling
> For VP and VE variants, we now evaluate quadratic and cosine schedules on superresolution. Using our paper’s notation:
> - VP: $\alpha_t=1-t^2$ (quadratic), $\alpha_t=\cos(\frac{\pi}{2}t)$ (cosine)
> - VE: $\beta_t=\sigma_{max}t^2$ (quadratic), $\beta_t=\sigma_{max}\sin(\frac{\pi}{2}t)$ (cosine)
> with $\gamma_t = \sigma_{max}^{-1}\beta_t$.
>
> #### SDB (VP)
> |Schedule|FID↓|LPIPS↓|PSNR↑|SSIM↑|
> |-|-|-|-|-|
> |Linear|87.08|0.228|**25.91**|**0.724**|
> |Quadratic|**75.25**|0.174|24.77|0.680|
> |Cosine|77.99|**0.160**|24.75|0.674|
>
> #### SDB (VE)
> |Schedule|FID↓|LPIPS↓|PSNR↑|SSIM↑|
> |-|-|-|-|-|
> |Linear|94.73|0.226|25.90|0.718|
> |Quadratic|76.88|**0.159**|25.29|0.678|
> |Cosine|**76.14**|0.217|**26.17**|**0.732**|
>
> Sophisticated schedules shift the perception–distortion trade-off. Linear favors distortion metrics; quadratic and cosine improve perceptual ones. Cosine SDB (VE) outperforms all methods in 3 of 4 metrics, setting a new SOTA. These results suggest deeper study of SDB’s dynamics is worthwhile.
> ## (Q5.) Coupling range and null spaces
> We appreciate the reviewer’s observation. While range and null spaces are orthogonal, their contributions to the SDE are separately defined by $\alpha_t, \beta_t, \gamma_t$, but can be functionally related, as in SDB (VP) and SDB (VE). SDB estimates the score via $\mathrm{E}[\mathbf{x}_0|\mathbf{x}_t]$, so the network inputs and outputs span both subspaces and implicitly learn the coupling, as the reviewer suggests—e.g., in inpainting, unmasked data guides masked estimation.
> ## Summary
> We hope these responses address all concerns and remain open to further questions.
>
> ---
> [1] Lipman et al., "Flow Matching for Generative Modeling", ICLR 2023.
> [2] Tivnan et al., "Fourier Diffusion Models", IEEE TMI, 2025.
> [3] Kamm et al., "Stochastic Magnus Expansion and SPDEs", J. Sci. Comput., 2021.
> [4] Bowen et al., "Subspace Diffusion Generative Models", ECCV 2022.
> [5] Nilsback & Zisserman, "Automated Flower Classification", ICVGIP 2008.
> [6] Kim et al., "Neural Schrödinger Bridge", ICLR 2024.
> [7] De Bortoli et al., "Schrödinger Bridge Flow", NeurIPS 2024.
> [8] Krizhevsky, "Learning Features from Tiny Images", 2009.

---

> ### Author Response · Authors · 2025-08-05
>
> Dear Reviewer FGzg,
>
> We are grateful for your initial review and the careful attention you’ve given our work. We welcome the opportunity to clarify any points of confusion and address any questions you may have.

---

> > ### Comment · Area_Chair_JVk1 · 2025-08-05
> >
> > Dear reviewer FGzg,
> >
> > As the discussion phase is reaching an end, it is important that you engage in a discussion with the authors. The authors made some efforts to reply to your comments. Did the answer make you strengthen or reconsider your decision? In any case, please explain why in a comment and your thoughts about the other reviews.
> >
> > Regards,
> >
> > The AC

---

### Official Review · Reviewer_Tpm3 · 2025-07-01

**Clarity:** 3
**Significance:** 3
**Originality:** 3
**Rating:** 4
**Confidence:** 4

**Summary:**

This work proposes System-embedded Diffusion Bridge Models (SDBs) that embed the known linear measurement system into the coefficients of the SDE process for solving linear inverse problems. Experiments show the effectiveness of the proposed method.

**Questions:**

1. The experiments, while covering standard benchmarks, are all 2D, moderately sized, and with relatively controlled corruptions. It would strengthen the work to see results on more diverse and challenging real-world scenarios (e.g., more realistic noise, higher noise levels, extreme undersampling, or out-of-distribution corruptions).

**Ethical Concerns:**

["NO or VERY MINOR ethics concerns only"]

**Final Justification:**

Considering the authors' rebuttal that solved most of my concerns, I keep my positive score.

**Limitations:**

The authors have provided the limitations of their work.

**Quality:**

3

**Strengths And Weaknesses:**

**Strength:**
1. This work provides a theoretically grounded method for embedding measurement system into the inverse process, filling the gap of prior bridge methods;
2. The proposed SDB is shown to be robust when the measurement system differs between training and deployment;
3. Experiments shown the effectiveness of the proposed method across diverse domains, including natural images and medical images;

**Weaknesses:**
1. While the paper includes some evaluations for system parameter changes, these are controlled perturbations. It would be valuable to explore settings where the operator A or noise model are misspecified in more adversarial or realistic ways (e.g., using a wrong noise distribution, or a blurred operator);
2. As the proposed SDB requires paired data for supervised training, which may not always be available, especially for rare measurements or privacy-sensitive domains. It does not address scenarios where only unpaired data is accessible, limiting applicability compared to plug-and-play or unsupervised diffusion approaches.
3. There is little ablation on key design choices such as the choice of noise schedules, the effect of imperfect A, or the benefit of each SDB component.
4. Some theoretically motivated variants (SDB-SB, SDB-VP, SDB-VE) are included, but the empirical results do not fully disentangle when each variant is preferable. For instance, SDB-SB generally performs best, but it is not entirely clear under what circumstances SDB-VP/VE might be better or whether their variance schedules are optimal for the tasks at hand.

---

> ### Author Rebuttal · Authors · 2025-07-31
>
> We thank the reviewer for the insightful comments and appreciation of our work. We address the mentioned weaknesses and questions below.
>
> ## (W1.) More challenging misspecified settings
>
> We agree with the reviewer that, although the system parameter changes we presented correspond to realistic scenarios, they represent controlled perturbations. To address this, we evaluate four additional settings that are either more adversarial or more reflective of practical degradations.
>
> ### Misspecified operator
>
> First, we evaluate all supervised methods under a drastically modified operator in the superresolution task. We increase the downsampling kernel size from the default $4\times4$ to $32\times32$, which yields inputs sharing very little mid- and high-frequency content with the training-time pseudoinverse reconstructions (PRs). While the performance gap between methods naturally shrinks due to the increased difficulty, SDB remains the best-performing method in 3 out of 4 metrics.
>
> |Method|FID↓|LPIPS↓|PSNR↑|SSIM↑|
> |-|-|-|-|-|
> |IRSDE|130.47|0.684|17.85|0.381|
> |I2SB|131.33|**0.681**|17.84|0.381|
> |DDBM|131.55|0.700|17.83|0.380|
> |GOUB|134.66|0.690|17.85|0.370|
> |SDB (SB)|**129.87**|0.682|**17.86**|**0.390**|
>
> Next, we replace the downsampling operator with JPEG compression preserving only 30% of the original image quality, simulating high-frequency loss via a markedly different mechanism. The results replicate our original ranking, with SDB (SB) outperforming all others in 3 of 4 metrics.
>
> |Method|FID↓|LPIPS↓|PSNR↑|SSIM↑|
> |-|-|-|-|-|
> |IRSDE|99.43|**0.283**|24.81|0.651|
> |GOUB|101.08|0.291|24.91|0.662|
> |I2SB|93.62|0.322|25.12|0.668|
> |DDBM|92.39|0.373|25.19|0.685|
> |SDB (SB)|**90.26**|0.380|**25.41**|**0.692**|
>
> ### Misspecified noise model
>
> We also consider a misspecified noise setting, replacing the Gaussian noise with Poisson noise to simulate photon counting physics. Specifically, given a signal sample $\mathbf{x}$, measurements $\mathbf{y}$ are drawn from $\text{Poisson}(\mathbf{I}_0 \cdot \exp(-\mathbf{A}\mathbf{x}))$, where $\mathbf{I}_0$ is the incident photon intensity. We evaluate on the CT reconstruction task, where this noise model closely reflects X-ray acquisition physics. Despite never explicitly observing Poisson noise and relying only on its Gaussian approximation in the limit, SDB outperforms all supervised baselines across different intensity values.
>
> |Method|$\mathbf{I}_0=10000$||| |$\mathbf{I}_0=5000$||| |$\mathbf{I}_0=1000$||| |
> |-|-|-|-|-|-|-|-|-|-|-|-|-|
> | |FID↓|LPIPS↓|PSNR↑|SSIM↑|FID↓|LPIPS↓|PSNR↑|SSIM↑|FID↓|LPIPS↓|PSNR↑|SSIM↑|
> |GOUB|17.78|0.0230|44.85|0.9717|17.83|0.0240|44.59|0.9705|20.52|0.0491|42.79|0.9588|
> |IRSDE|17.29|0.0256|44.17|0.9681|17.86|0.0299|43.88|0.9665|21.12|0.0678|41.84|0.9506|
> |DDBM|20.69|0.0402|44.11|0.9622|20.57|0.0406|43.93|0.9613|21.75|0.0575|42.53|0.9525|
> |I2SB|21.91|0.1014|41.68|0.9242|21.82|0.1018|41.54|0.9231|22.43|0.1170|40.44|0.9130|
> |SDB (SB)|**14.03**|**0.0193**|**46.05**|**0.9799**|**14.61**|**0.0203**|**45.70**|**0.9785**|**19.80**|**0.0461**|**43.44**|**0.9657**|
>
> To propose a more adversarial setting, we also replace the Gaussian noise with Poisson noise in the MRI reconstruction task, where Poisson noise is typically unnatural, thus imposing a highly out-of-distribution challenge. SDB again outperforms all methods in 3 of 4 metrics across all intensities, while nearly matching the best SSIM.
>
> |Method|$\mathbf{I}_0=20.0$| | | |$\mathbf{I}_0=16.0$| | | |$\mathbf{I}_0=12.0$| | | |$\mathbf{I}_0=8.0$| | | |
> |-|-|-|-|-|-|-|-|-|-|-|-|-|-|-|-|-|
> | |FID↓|LPIPS↓|PSNR↑|SSIM↑|FID↓|LPIPS↓|PSNR↑|SSIM↑|FID↓|LPIPS↓|PSNR↑|SSIM↑|FID↓|LPIPS↓|PSNR↑|SSIM↑|
> |DDBM|40.22|0.265|21.95|**0.557**|39.09|0.236|22.69|**0.592**|38.30|0.204|23.61|**0.635**|36.67|0.165|24.78|**0.687**|
> |GOUB|39.83|0.282|21.50|0.525|39.33|0.253|22.21|0.558|38.41|0.220|23.08|0.599|36.95|0.179|24.22|0.652|
> |IRSDE|39.85|0.271|21.58|0.538|39.41|0.244|22.27|0.572|38.42|0.214|23.14|0.614|37.07|0.177|24.30|0.668|
> |I2SB|40.04|0.277|21.45|0.522|39.06|0.249|22.17|0.555|38.13|0.217|23.08|0.596|37.12|0.178|24.26|0.649|
> |SDB (SB)|**39.73**|**0.251**|**22.07**|**0.557**|**38.97**|**0.225**|**22.82**|0.589|**38.15**|**0.194**|**23.76**|0.629|**36.63**|**0.157**|**24.97**|0.681|
>
> ## (W2.) Limitation to paired data
>
> We thank the reviewer for highlighting this important limitation. We agree that unpaired data scenarios are critical in many practical applications, where paired-data methods, including SDB, are not currently applicable. This limitation fundamentally distinguishes paired-data supervised approaches from plug-and-play or unsupervised methods, which trade off some performance for broader applicability. We will include a discussion of this limitation and potential directions in the updated manuscript.
>
> ## (W3.) Additional ablation studies
>
> The original evaluation of misspecified measurement systems was broad, covering two parameters for both the CT and MRI tasks (see Figure 3 for MRI and Figure 4 (Appendix) for CT). In general, we ensured the evaluation focused entirely on i) benefits of replacing scalar SDEs with matrix-valued ones, and ii) differentiating the effect of noise scheduling. To isolate these factors, all baselines (both un- and supervised) were retrained from scratch using the same architecture, training procedures, and optimization hyperparameters—varying only the SDE formulation.
>
> Regarding noise schedule design, Appendix E.2 includes a broad hyperparameter analysis for SDB (SB). Since SB schedule fixes a parameterization, we verify which "shape" performs best by grid-searching over $b_0$, $b_1$, and truncation points $\epsilon_1$, $\epsilon_2$.
>
> ### Ablation studies with linear, quadratic and cosine scheduling
>
> We agree the schedule choice for VP and VE variants was unaddressed. We thus test linear, quadratic, and cosine schedulings. Using our notation: $\alpha_t = 1 - t^2$ (quadratic SDB (VP)), $\alpha_t = \cos\left(\frac{\pi}{2} t\right)$ (cosine SDB (VP)), $\beta_t = \sigma_{\max} t^2$ (quadratic SDB (VE)), and $\beta_t = \sigma_{\max} \sin\left(\frac{\pi}{2} t\right)$ (cosine SDB (VE)). Other coefficients follow lines 202–203 of our submission, assuming $\gamma_t = \sigma_{\max}^{-1} \beta_t$ for SDB (VE).
>
> #### SDB (VP)
>
> |Schedule|FID↓|LPIPS↓|PSNR↑|SSIM↑|
> |-|-|-|-|-|
> |Linear|87.08|0.228|**25.91**|**0.724**|
> |Quadratic|**75.25**|0.174|24.77|0.680|
> |Cosine|77.99|**0.160**|24.75|0.674|
>
> #### SDB (VE)
>
> |Schedule|FID↓|LPIPS↓|PSNR↑|SSIM↑|
> |-|-|-|-|-|
> |Linear|94.73|0.226|25.90|0.718|
> |Quadratic|76.88|**0.159**|25.29|0.678|
> |Cosine|**76.14**|0.217|**26.17**|**0.732**|
>
> More sophisticated schedulers yield a trade-off between perception (FID, LPIPS) and distortion (PSNR, SSIM). Linear favors distortion; quadratic and cosine sacrifice distortion for perceptual quality. Interestingly, with Cosine SDB (VE), we set a new state-of-the-art on 3 of 4 metrics when compared with the initial baselines, showing that deeper exploration of SDB dynamics is promising.
>
> ## (W4.) Preference over SDB variants
>
> We thank the reviewer for noting the need to clarify preference among SDB variants. SDB (SB) is preferred, based on theory (Theorem 2) and experiments (Table 2). For VP and VE, we suggest prioritizing VP for performance, supported by its ranking over VE in Table 2. Visual inspection indicates VE yields increased diversity, matching literature on unconditional diffusion models.
>
> The ablation above indicates a cosine schedule for VE and linear for VP is preferred. Appendix E.2’s ablation over SB shapes confirms that, for a given reconstruction task, choosing an optimal schedule is a complex and challenging problem.
>
> ## (Q1.) More diverse and real-world scenarios
>
> We hope that the new results on extreme misspecifications above already showcase SDB’s robustness. We add two new benchmarks for other real-world scenarios.
>
> ### Motion blur
>
> To test SDB on an ill-conditioned, rank-deficient system, we benchmark motion blur on 128×128 Flowers102 images [1], per Reviewer unSw’s suggestion. The system matrix is a block Toeplitz with Toeplitz blocks (BTTB), implemented as 2D convolutions. For pseudoinverse, zero-order Tikhonov regularization (Wiener filtering) is applied in frequency domain. Though it must handle the approximate pseudoinverse application, SDB matches the best LPIPS and outperforms supervised baselines on other metrics, as shown below:
>
> |Method|FID↓|LPIPS↓|PSNR↑|SSIM↑|
> |-|-|-|-|-|
> |GOUB|17.03|0.053|23.84|0.813|
> |IRSDE|16.27|0.033|26.76|0.842|
> |DDBM|16.00|0.040|29.74|0.892|
> |I2SB|15.62|**0.025**|29.29|0.874|
> |SDB (SB)|**14.89**|**0.025**|**30.39**|**0.903**|
>
> ### Nonlinear contrast
>
> To show SDB’s applicability to nonlinear systems, we benchmark CIFAR10 [2] with a nonlinear contrast system $\mathbf{A}(\mathbf{x}) = \sigma(k(\mathbf{x} - \alpha))$, where $\sigma$ is sigmoid, $k$ controls contrast, and $\alpha$ is bias. We apply the linearization method from response **(W2., Q4.)**. The table shows SDB once again outperforming supervised baselines:
>
> |Method|FID↓|LPIPS↓|PSNR↑|SSIM↑|
> |-|-|-|-|-|
> |I2SB|2.97|0.00103|37.53|0.9906|
> |GOUB|4.60|0.00182|32.54|0.9743|
> |DDBM|1.92|0.00016|42.74|0.9963|
> |IRSDE|4.35|0.00140|33.53|0.9789|
> |SDB (SB)|**1.31**|**0.00015**|**44.48**|**0.9977**|
>
> ## Summary
>
> We hope these comments address the reviewer’s concerns and are open to further discussion.
>
> ---
>
> [1] Nilsback, M-E. and Zisserman, A., "Automated flower classification over a large number of classes", Proceedings of the Indian Conference on Computer Vision, Graphics and Image Processing, 2008.
>
> [2] Krizhevsky, A., "Learning Multiple Layers of Features from Tiny Images", 2009.

---

> > ### Comment · Reviewer_Tpm3 · 2025-08-08
> > **Official Comment by Reviewer Tpm3**
> >
> > Thanks for the authors' rebuttal. As all the concerns are solved, I will maintain my positive score.

---

> > > ### Author Response · Authors · 2025-08-08
> > >
> > > Dear Reviewer Tpm3,
> > >
> > > Thank you for your thoughtful participation in both the review and discussion phases. We would be glad to address any remaining concerns that might prompt you to consider raising your score. Thank you in advance for your consideration.

---

> ### Author Response · Authors · 2025-08-05
>
> Dear Reviewer Tpm3,
>
> Thank you kindly for your initial assessment and the constructive feedback provided. We would be glad to engage further, address any outstanding questions, and ensure that all aspects of the work are clear and well-supported.

---

> > ### Comment · Area_Chair_JVk1 · 2025-08-05
> >
> > Dear reviewer Tpm3,
> >
> > As the discussion phase is reaching an end, it is important that you engage in a discussion with the authors. The authors made some efforts to reply to your comments. Did the answer make you strengthen or reconsider your decision? In any case, please explain why in a comment and your thoughts about the other reviews.
> >
> > Regards,
> >
> > The AC

---

### Official Review · Reviewer_unSw · 2025-07-03

**Clarity:** 3
**Significance:** 3
**Originality:** 3
**Rating:** 5
**Confidence:** 4

**Summary:**

The paper presents a supervised framework for solving linear inverse problems using score-based generative models. The core motivation is to embedding known information about the linear measurement system into the diffusion SDEs, which contrasts with previous diffusion bridge methods that often remain agnostic to this structural information. Experimental results demonstrate the effectiveness of the proposed method.

**Questions:**

Please refer to the section of Weaknesses.

**Ethical Concerns:**

["NO or VERY MINOR ethics concerns only"]

**Final Justification:**

The authors have clarified the concerns in the responses, including the proof demonstrating that the proposed method can achieve principled posterior sampling. Thus, I raise my rating.

**Limitations:**

Yes.

**Quality:**

3

**Strengths And Weaknesses:**

Strengths:

1. The writing is clear and the motivations are promising.

2. The construction of the system-embedded SDEs is reasonable and novel.

3. Experimental results demonstrate the effectiveness of embedding measurement-system into diffusion SDEs.

Weaknesses:

1. The method relies on pseudo-inverse of the forward measurement matrix A. In the experiments the author use SVD to compute the pseudo-inverse.  However, for some linear measurement system, SVD is hard, e.g., motion deburing [R1]. This could limits the practicality of the proposed method.

2. I believe that the conventional datasets for diffusion-based solvers (ImageNet, FFHQ) [R1，R2] should be included to better demonstrate the effectiveness of the proposed method.

3. It is not immediately clear how the proposed model ensures sampling from the correct conditional distribution $p(x|y)$. The authors should provide further clarification or theoretical justification on this point.

Reference:

[R1] Chung, Hyungjin, et al. "Diffusion posterior sampling for general noisy inverse problems." arXiv preprint arXiv:2209.14687 (2022).

[R2] Zhang, Bingliang, et al. "Improving diffusion inverse problem solving with decoupled noise annealing." Proceedings of the Computer Vision and Pattern Recognition Conference. 2025.

---

> ### Author Rebuttal · Authors · 2025-07-31
>
> We thank the reviewer for the insightful feedback and for appreciating our work. To address the noted weaknesses, we provide separate comments below.
>
> ## (W1.) Limited practicality due to SVD computation
>
> We appreciate this observation. In the main manuscript, we aimed to provide an intuitive view of the pseudoinverse via its relationship with SVD, where non-zero singular values are replaced with their reciprocals. However, SDB does not require explicit SVD or matrix storage. Instead, we rely on how $\mathbf{A}$, $\mathbf{A}^\top$, and $\mathbf{A}^+$ act on inputs. For instance, in the superresolution task, we do not store the downsampling matrix but apply $\mathbf{A}$ via 2D average pooling. This allows for implementing $\mathbf{A}^+$ as nearest-neighbor up-sampling, which is much more efficient than storing the dense SVD.
>
> ### Motion deblurring task
>
> We agree that numerical instabilities may affect SDB’s practicality, including SVD if used. To address this, we consider motion deblurring on 128×128 Flowers102 images [1]. The system matrix has a block Toeplitz structure with Toeplitz blocks (BTTB), implemented using 2D convolutions. For pseudoinverse approximation, we apply zero-order Tikhonov regularization (Wiener filtering) in the frequency domain. While approximate, SDB matches the best LPIPS and clearly outperforms all supervised baselines on the remaining metrics, as shown below.
>
> |Method|FID↓|LPIPS↓|PSNR↑|SSIM↑|
> |-|-|-|-|-|
> |GOUB|17.03|0.053|23.84|0.813|
> |IRSDE|16.27|0.033|26.76|0.842|
> |DDBM|16.00|0.040|29.74|0.892|
> |I2SB|15.62|**0.025**|29.29|0.874|
> |SDB (SB)|**14.89**|**0.025**|**30.39**|**0.903**|
>
> ### Further analysis of stability
>
> We note that CT reconstruction is the only case where we use a precomputed SVD of the system matrix. In this setting, truncated SVD serves as a regularization technique by discarding small singular values, typically tied to high-frequency components and noise. In our paper, the parameter $\tau$ controls this regularization. During training, we set $\tau=3.2$ to emulate a cut-off filter, zeroing out the lowest singular values which typically cause stability issues. Nonetheless, as Table 2 in the main paper shows, SDB consistently outperforms all baselines, indicating robustness to such regularization. This is further supported by the misspecified CT setting (Appendix E.1, Figure 4), where increasing $\tau$ leads SDB to maintain or even expand its advantage.
>
> ### Summary
>
> We hope the two discussed experiments clarify concerns about SVD-related computations and their influence on SDB’s performance.
>
> ## (W2.) Conventional datasets for diffusion-based solvers
>
> We thank the reviewer for this suggestion. Regarding FFHQ [2], we find it highly similar to the CelebA-HQ dataset used in our work. Both feature high-resolution, centered human faces, with FFHQ being about twice as large (70,000 samples). As for ImageNet [3], we fully agree it would be a strong testbed for diffusion bridges. However, its scale makes training from scratch extremely costly. Even recent SOTA bridge methods for both paired (I2SB [4], IR-SDE, GOUB, DBBM) and unpaired (UNSB [5], SBF [6]) data rarely use ImageNet. I2SB is a notable exception — despite NVIDIA support and initialization from a pretrained unconditional ImageNet diffusion model, its authors recommend continued training with 2 nodes of 8×32GB V100 GPUs, highlighting the significant computational burden.
>
> ## (W3.) Correct sampling from $p(x|y)$
>
> Thank you for this comment. To begin, it is important to clarify that many other diffusion bridge methods such as I2SB, GOUB, or DDBM, use a non-Markovian stochastic process, where $\mathbf{x}_t$ is a function of both $\mathbf{x}_0$ and $\mathbf{x}_1$, whereas our method uses a Markovian process, that is, a standard forward diffusion process, where $p(\mathbf{x}_t|\mathbf{x}_0)$ is well-defined. More specifically, our method is a special case of score-based generative models, using the general formulation with matrix-valued drift and diffusion coefficients in Appendix A of [8]. With this in mind, we provide below a formal theorem and proof showing that SDB is a principled posterior sampler of $p(\mathbf{x}|\mathbf{y})$.
>
> ### Theorem 3
> Given a forward measurement model $p(\mathbf{y}|\mathbf{x}) = \mathcal{N}(\mathbf{A} \mathbf{x} , \Sigma)$, one can generate principled posterior samples from $p(\mathbf{x}|\mathbf{y})$, using the reverse-time stochastic process of SDB, where the forward stochastic process as defined by Eqs. (1), (7), (8a), and (8b), parameterized by the time-dependent scalar coefficients, $\gamma_t$, $\alpha_t$, and $\beta_t$, satisfy $\lim_{t\rightarrow 1} \gamma_t = 1$ and $\lim_{t\rightarrow 1} \frac{\alpha_t^2}{\beta_t} = 0$. Here "principled" means the training and sampling methodologies result in an asymptotically exact probabilistic model in the limit of infinite training data and infinite model capacity.
>
> ### Lemma 3.1
> The pseudoinverse reconstruction $\mathbf{A}^{+}\mathbf{y}$ is a sufficient statistic for estimation of $\mathbf{x}$ given $\mathbf{y}$. That is, $p(\mathbf{x} | \mathbf{y})= p(\mathbf{x} | \mathbf{A}^{+} \mathbf{y})$. This is true because 1) the range space component of the measurements can be exactly recovered by re-applying $\mathbf{A}$ to $\mathbf{A}^{+} \mathbf{y}$, and 2) the null space component contains no useful information about $\mathbf{x}$.
>
> ### Lemma 3.2
> A sample from $p(\mathbf{z}|\mathbf{y}) = \mathcal{N}(\mathbf{A}^{+}\mathbf{y},  \beta_1 (\mathbf{I} - \mathbf{A}^{+} \mathbf{A}))$ is also a sufficient statistic for estimation of $\mathbf{x}$ given $\mathbf{y}$. That is, $p(\mathbf{x} | \mathbf{y})= p(\mathbf{x} | \mathbf{z})$. This is true because the null-space component of the pseudoinverse reconstruction is zero. Therefore, there is no information in the null-space component that is useful for estimation of $\mathbf{x}$ and so the additive null-space noise preserves sufficiency for estimation of $p(\mathbf{x}|\mathbf{y})$.
>
> ### Lemma 3.3
> The SDB forward stochastic process at the final time step, $p(\mathbf{x}_1|\mathbf{x}_0=\mathbf{x})$,  as defined by (1), (7), (8a), and (8b) is identically distributed to the SDB initializer, $p(\mathbf{z}|\mathbf{x})$  from Lemma 3.2., assuming
>
> $\lim_{t\rightarrow 1} \gamma_t = 1$ and $\lim_{t\rightarrow 1} \frac{\alpha_t^2}{\beta_t} = 0$.
>
> This is true because one can show they are both conditional Gaussian random vectors with the same mean, $\mathbf{A}^{+} \mathbf{A} \mathbf{x}$, and same covariance, $\mathbf{A}^{+} \Sigma {\mathbf{A}^{+}}^T +  \beta_1 (\mathbf{I} - \mathbf{A}^{+} \mathbf{A})$.
>
> ### Lemma 3.4
> The trained SDB model is a principled sampler of $p(\mathbf{x}_0 | \mathbf{x}_1)$.  This is valid because the SDB is a subset of score-based generative models using a standard Markovian forward diffusion stochastic process.  According to Anderson theorem [7] and the general framework in Appendix A of [8], the reverse SDE in [8] using a learned score estimator in (3) results in a principled sampler of $p(\mathbf{x}_0|\mathbf{x}_1)$. To be clear, the score network is not the exact score function, but the methodology is "principled" because it approaches the exact score with large amounts of training data and many parameters in the score network.
>
> ### Proof of Theorem 3:
> The following procedure results in principled posterior samples of $p(\mathbf{x}|\mathbf{y})$:
> 1) Compute the pseudoinverse reconstruction $\mathbf{A}^{+}\mathbf{y}$
> 2) Add null-space noise to sample $p(\mathbf{z}|\mathbf{y}) = \mathcal{N}(\mathbf{z}; \mathbf{A}^{+}\mathbf{y},  \beta_1 (\mathbf{I} - \mathbf{A}^{+} \mathbf{A}))$
> 3) Sample the SDB reverse process $p(\mathbf{x}_0=\mathbf{x}  | \mathbf{x}_1 = \mathbf{z})$
>
> According to Lemma 3.1 and 3.2, step 2 results in a sufficient statistic, $\mathbf{z}$, for estimation of $p(\mathbf{x}|\mathbf{y})$. According to Lemma 3.3, the initialization $\mathbf{x}_1=\mathbf{z}$ is valid because the SDB initializer, $p(\mathbf{z}|\mathbf{x})$, is identically distributed to the forward process, $p(\mathbf{x}_1 | \mathbf{x}_0=\mathbf{x})$, assuming $\mathbf{x}_0$ is sampled from $p(\mathbf{x})$ during training. According to Lemma 3.4, the trained SDB model is a principled posterior sampler of $p(\mathbf{x}_0=\mathbf{x} | \mathbf{x}_1=\mathbf{z})$. Therefore, the result will be principled posterior samples of $p(\mathbf{x}|\mathbf{y})$.
>
> ## Summary
>
> We hope that the provided comments clarify the mentioned issues. We are open to further discussion if any further questions arise.
>
> ---
>
> [1] Nilsback, M-E. and Zisserman, A., "Automated flower classification over a large number of classes",  Proceedings of the Indian Conference on Computer Vision, Graphics and Image Processing, 2008.
>
> [2] Zhang, B., et al. "Improving diffusion inverse problem solving with decoupled noise annealing", Proceedings of the Computer Vision and Pattern Recognition Conference. 2025.
>
> [3] Chung, H., et al. "Diffusion posterior sampling for general noisy inverse problems", International Conference on Learning Representations, 2023.
>
> [4] Liu, G.H., et al., "I2SB: Image-to-Image Schrödinger Bridge", International Conference on Machine Learning, 2023.
>
> [5] Kim, B., et al., "Unpaired Image-to-Image Translation via Neural Schrödinger Bridge", International Conference on Learning Representations, 2024.
>
> [6] De Bortoli, V., et al., "Schrödinger Bridge Flow for Unpaired Data Translation", Neural Information Processing Systems, 2024.
>
> [7] Anderson, Brian DO. "Reverse-time diffusion equation models." Stochastic Processes and their Applications 12.3 (1982): 313-326.
>
> [8] Song, Yang, et al. "Score-based generative modeling through stochastic differential equations." arXiv preprint arXiv:2011.13456 (2020).

---

> > ### Comment · Reviewer_unSw · 2025-08-07
> >
> > I appreciate the authors for their thoughtful responses, especially for providing the proof demonstrating that the proposed method can achieve principled posterior sampling. I will raise my rating accordingly.

---

> ### Author Response · Authors · 2025-08-05
>
> Dear Reviewer unSw,
>
> We sincerely appreciate your initial evaluation and thoughtful comments. We would be happy to engage in further discussion, resolve any potential issues or concerns, and clarify any points that may require elaboration.

---

> > ### Comment · Area_Chair_JVk1 · 2025-08-05
> >
> > Dear reviewer unSw,
> >
> > As the discussion phase is reaching an end, it is important that you engage in a discussion with the authors. The authors made some efforts to reply to your comments. Did the answer make you strengthen or reconsider your decision? In any case, please explain why in a comment and your thoughts about the other reviews.
> >
> >
> > Regards,
> >
> > The AC

---

> ### Author Response · Authors · 2025-08-08
>
> Dear Reviewer unSw,
>
> We are deeply grateful for the insightful feedback provided during the review process and truly appreciate the decision to raise the rating.

---

### Note · Authors · 2025-08-11

Dear Reviewers,

Thank you for the valuable feedback on our submission. Below, we summarize our rebuttal, highlighting how each issue raised in the reviews has been resolved.

### 1. Two new benchmarks:

a. motion deblurring on Flowers102 with Tikhonov regularization for pseudoinverse computation.

b. nonlinear contrast on CIFAR10 with local linearization.

These address unSw and FGzg’s concerns on unstable/costly SVDs and FGzg’s on nonlinear scenarios. In both cases, SDB outperforms prior SOTA.

### 2. Four new evaluations under a more challenging misspecified setting:

a. misspecified system response matrix:
- extreme downsampling (scale 32) on DIV2k
- JPEG compression (very low quality) instead of downsampling on DIV2K

b. misspecified noise:
- Poisson instead of Gaussian on MRI reconstruction (adversarial)
- Poisson instead of Gaussian on CT reconstruction (realistic)

These address Tpm3’s concerns on more challenging system parameter changes. SDB again outperforms prior SOTA.

### 3. Four new theoretical results:

a. proof of SDB as a principled posterior sampler (unSw)

b. justification of the Magnus expansion’s generality (FGzg)

c. SDB with a locally linearized nonlinear system (FGzg)

d. complexity of a single forward/reverse step and the full reverse process (FGzg)

### 4. Two additional ablation studies:

a. comparison of linear, quadratic and cosine noise schedules for SDB (VP) and SDB (VE)

b. runtime analysis of overhead from the matrix-valued formulation

Here, a. provides ablations mentioned by Tpm3 and FGzg, while b. shows SDB's overhead is negligible (FGzg).

### 5. Two new baselines based on novel optimal transport methods for unpaired data:

a. Unpaired Image-to-Image Translation via Neural Schrödinger Bridge

b. Schrödinger Bridge Flow for Unpaired Data Translation

These were proposed by FGzg.

### 6. New discussions on:

a. why SDB is preferred over methods not conditioned by the system information (JgTN)

b. SDB coupling information from the null and range spaces (FGzg)

c. preference over SDB variants (Tpm3)

d. perception-distortion tradeoff (JgTN)

e. several degradations handled by a single model (JgTN)

We hope these additions, together with the positive initial feedback, provide convincing evidence of the novelty of our work. We kindly ask you to raise your rating if your issues are resolved. As a final note, we thank reviewer unSw for deciding to raise their score, and kindly ask to modify the rating in the review.

---

### Decision · Program_Chairs · 2025-09-17

**Decision:**

Accept (poster)

**Comment:**

This paper introduces System-embedded Diffusion Bridge Models (SDBs), a framework for supervised diffusion bridge methods that explicitly embeds the known linear measurement system into the coefficients of matrix-valued stochastic differential equations. The key contribution lies in leveraging the range-nullspace decomposition of linear operators to design specialized diffusion processes that handle range and null space components differently. The authors propose three variants (SDB-SB, SDB-VP, SDB-VE) and demonstrate improvements over existing bridge methods (I2SB, IR-SDE, GOUB, DDBM) across four inverse problems: image inpainting, super-resolution, CT reconstruction, and MRI reconstruction.


During the rebuttal, the authors provided additional clarifications and results to address the reviewers remarks ( comparison with SoTA baselines, results on more challenging misspecified settings (reviewer Tpm3), ablation studies influence of pseudo-inverse stability (FGzg) ).  The responses and promised clarifications have addressed the reviewers points and should be included in a revised version.